# Autophagy lipidation machinery regulates axonal microtubule dynamics but is dispensable for survival of mammalian neurons

A. Negrete-Hurtado[1,5], M. Overhoff[1,5], S. Bera[1,5], E. De Bruyckere[1], K. Schätzmüller[1], M.J. Kye [2], C. Qin[3], M. Lammers [3], V. Kondylis [1], I. Neundorf [4] & N.L. Kononenko[1✉]

Neurons maintain axonal homeostasis via employing a unique organization of the microtubule (MT) cytoskeleton, which supports axonal morphology and provides tracks for intracellular transport. Abnormal MT-based trafficking hallmarks the pathology of neurodegenerative diseases, but the exact mechanism regulating MT dynamics in axons remains enigmatic. Here we report on a regulation of MT dynamics by AuTophaGy(ATG)-related proteins, which previously have been linked to the autophagy pathway. We find that ATG proteins required for LC3 lipid conjugation are dispensable for survival of excitatory neurons and instead regulate MT stability via controlling the abundance of the MT-binding protein CLASP2. This function of ATGs is independent of their role in autophagy and requires the active zone protein ELKS1. Our results highlight a non-canonical role of ATG proteins in neurons and suggest that pharmacological activation of autophagy may not only promote the degradation of cytoplasmic material, but also impair axonal integrity via altering MT stability.

---

[1] CECAD Cluster of Excellence, Institute for Genetics, University of Cologne, Cologne, Germany. [2] Institute of Human Genetics, University Hospital Cologne, Cologne, Germany. [3] Institute of Biochemistry, Synthetic and Structural Biochemistry, University of Greifswald, Greifswald, Germany. [4] Department of Chemistry, Institute of Biochemistry, University of Cologne, Cologne, Germany. [5] These authors contributed equally: A. Negrete-Hurtado, M. Overhoff, S. Bera. ✉email: n.kononenko@uni-koeln.de

Microtubules (MTs) play a fundamental role in the maintenance of axonal homeostasis by preserving axonal morphology and providing tracks for protein and organelle transport[1]. Unlike dendritic MTs, which exhibit a bidirectional polarity, axonal MTs are composed of predominantly plus-end-out oriented multimers of α-tubulin and β-tubulin heterodimers[2]. Although the majority of axonal MTs are more stable than the MTs of dendrites, axons also possess a fraction of shorter, dynamic MTs that undergo phases of growing (polymerization) and shrinking (depolymerization), a process known as dynamic instability[3]. This dynamic property of MTs is regulated by the intrinsic GTPase activity of tubulin, their association with MT-associated proteins (MAPs) and motor proteins[4], as well as by diverse MT posttranslational modifications (PTM), including acetylation, tyrosination and detyrosination[5].

The function of stable MTs in neurons is well studied and includes their pivotal role in providing tracks for long-distance transport[1]. In contrast, the purpose of dynamic MTs in otherwise extremely polarized neurons is not completely understood. It is believed that dynamic MTs act as sensors of cellular microenvironment, allowing the rapid reorganization of the cytoskeleton[6], required for synaptic plasticity and axonal regeneration[7,8]. Furthermore, since most axons are lifelong sustained, continued MT turnover is needed to prevent senescence through mechanical wear and to remove irreversible PTMs[9]. Given these prominent roles of MTs, it is not surprising that the tight regulation of stable versus dynamic equilibrium of MTs is pivotal for neuronal function. Dysregulation of MT dynamics is implicated in the majority of neurodegenerative disorders[10]. An increase in MT stability also correlates with decreased neuronal plasticity, which is known to occur during aging[11]. Although understanding the fine regulation of MT dynamics is crucial to unravel the etiology of neurodegeneration, the precise mechanism that regulates the equilibrium between dynamic and stable MTs in axons is currently unknown.

Axonal degeneration is also associated with aberrant autophagic activity. Autophagy is an evolutionary conserved process that provides nutrients during starvation and eliminates defective proteins and organelles via lysosomal degradation[12]. The most prevalent form of autophagy is macroautophagy, and during this process, portions of the cytoplasm are sequestered within double-membraned vesicles termed autophagosomes. These undergo subsequent maturation steps before being delivered to the lysosomes for degradation. Autophagy plays important roles in maintaining cellular homeostasis[13], and its dysfunction causes neurodegeneration characterized by axonal pathology[14] and accumulation of insoluble ubiquitin (Ub)-positive and Sequestosome-1 (SQSTM1, p62)-positive protein inclusions[15],. Intriguingly, prevention of inclusion formation by p62 ablation does not suppress neurodegeneration in autophagy-deficient mice[16], challenging a toxic role for inclusions in neurons and raising the question of precise physiological function of autophagy in the brain.

Autophagy induction depends on the activity of the ATG1/ULK1 and PI3KC3 complex I, while autophagosome formation requires two ubiquitin (Ub)-like conjugation machineries[13]. The Ub-like ATG12 conjugates with ATG5 (ATG12~ATG5), which further establishes a complex with ATG16L (ATG12~ATG5/ATG16L). E3-like activity of ATG12~ATG5/ATG16L catalyzes the lipid conjugation of Ub-like ATG8 family proteins, including MT-associated protein 1 light chain 3 (LC3) to autophagosomal membrane-resident phosphatidylethanolamine (PE)[17]. An important step for PE conjugation of ATG8 is its (pre-)processing by the cysteine protease ATG4, which exposes a C-terminal glycine residue essential for lipid conjugation[18]. Although canonically LC3 lipidation has been associated with the autophagy

pathway, recently this modification has been reported to occur during autophagy-unrelated processes[19]. For instance, in non-neuronal cells LC3 can be attached to the plasma membrane during virus infection[20] or it can be conjugated to endolysosomal membranes[21]. However, whether in neurons the LC3 lipidation machinery is exclusively involved in autophagy-associated protein degradation is currently unknown.

Here, we demonstrate that ATG proteins involved in lipid conjugation of LC3 are dispensable for survival of excitatory neurons and instead regulate the stability of axonal MTs independently of their role in canonical autophagy. Deleting ATG5 or ATG16L1 or increasing the availability of non-lipidated LC3 by using a mutant form of cysteine protease ATG4B causes the formation of axonal swellings due to impaired MT dynamics, a phenotype that hampers the transport of axonal cargo and compromises learning and memory-dependent neuronal activity. Our data demonstrate a function of the ATG lipidation machinery in neurons and suggest that modulation of autophagy activity in the brain may not only promote the degradation of bulk cytoplasmic material, but also impair axonal integrity via altering MT dynamics.

## Results

**Excitatory neurons survival does not require ATG5 or ATG16L1.** Since ablation of inclusion bodies in mice with impaired LC3 lipidation does not prevent neurodegeneration[16], we sought to investigate the exact mechanism by which the autophagy lipid conjugation machinery regulates neuronal function. To this end, we first eliminated the crucial autophagy component ATG5 in forebrain excitatory neurons by cross-breeding the *Atg5*flox/flox mice with *CamKIIα*-Cre mice[22] (Fig. 1a, Supplementary Fig. 1a). *Atg5*flox/flox:*CamKIIα*-Cre (ATG5 KO) mice were viable, but revealed a cessation of weight gain starting at about one month of age when compared to their WT littermates (Supplementary Fig. 1b). In agreement with published data[23], the amount of LC3 at autophagosomal membranes, monitored via LC3II (lipidated LC3) levels, was significantly downregulated (Fig. 1b, Supplementary Fig. 1c), while p62 was significantly increased in the cortex of mice lacking ATG5 (Supplementary Fig. 1e, f). Since autophagy is required for cellular survival[15] and apoptosis-related loss of neurons is described in several autophagy-deficient mouse models[24], we analyzed apoptotic signaling in forebrain excitatory ATG5 KO neurons. Surprisingly, the survival of neurons in the cortex and the hippocampus of 13-week-old mice was not affected by the ATG5 deletion (Fig. 1c–f). The lack of apoptosis in ATG5 KO brains was supported by the absence of cleaved caspase3 activity (Fig. 1g), a phenotype which was reproduced in an in-vitro cortico-hippocampal culture system, where the deletion of ATG5 was driven via a tamoxifen-dependent activation of the *CAG*-Cre promoter (Fig. 1h, i, Supplementary Fig. 1g–i). The survival-dispensable role of ATG5 in neurons was further confirmed by non-significant changes in the gene expression levels of pro-survival and pro-apoptotic genes (Fig. 1j). To elucidate whether this phenotype was a mere function of ATG5 or a more general phenomenon associated with a defective LC3 lipidation machinery, we deleted ATG16L1, another component of the LC3 lipid conjugation system (Fig. 1k, Supplementary Fig. 1j). In agreement with the data from the ATG5 KO mice, ATG16L1 was also dispensable for survival of cortical neurons in-vitro (Fig. 1l) and in-vivo (Fig. 1m, n).

**LC3 lipidation machinery maintains axonal homeostasis.** Even though ATG5 was dispensable for neuronal apoptotic signaling, ATG5 KO mice were characterized by increased postnatal

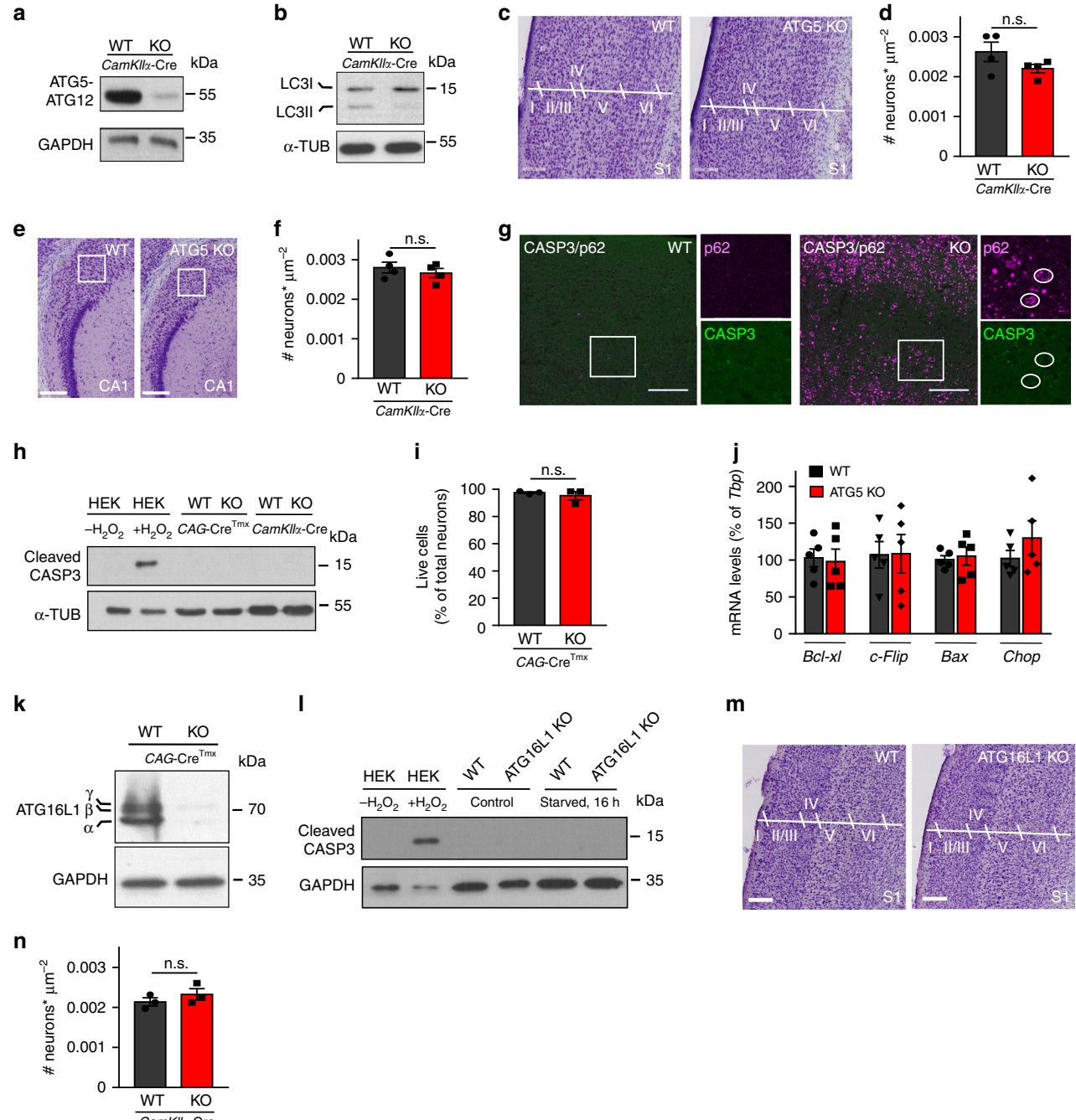

**Fig. 1 Survival of excitatory neurons is dispensable of ATG5 and ATG16L1. a** ATG5 protein levels in cortical brain lysates from 13-weeks-old WT and ATG5 KO mice. **b** LC3 lipidation is defective in ATG5 KO brains. **c–f** Histopathological analysis of Nissl-stained brain sections of ATG5 KO brains reveals unaltered number of cortical (**c**, **d**) (WT: 0.0026 ± 0.0002, KO: 0.0022 ± 0.0001. $p = 0.163$) and hippocampal neurons (**e**, **f**) (WT: 0.0028 ± 0.0003, KO: 0.0026 ± 0.0002. $p = 0.465$), $N = 4$ independent experiments each. Rectangular box in **e** depicts the hippocampal area used for quantification. Scale bars, 200 μm. **g** Representative confocal images of cortical brain sections from WT and ATG5 KO mice immunostained for cleaved Caspase-3 (CASP3, green) and co-immunostained for p62 (magenta). Rectangular boxes indicate areas magnified to the right. Circles in the KO indicate cells positive for p62, but negative for CASP3. Scale bars, 200 μm. **h** Immunoblot illustrating cleaved CASP3 levels in lysates from cultured neurons, as well as in cortical lysates from WT and ATG5 KO mice. HEK293T cells treated with $H_2O_2$ were used as a positive control. **i** Analysis of cellular viability using the Live/Dead cell assay in WT and ATG5 KO cultured neurons (WT: 97.55 ± 0.60%, KO: 95.25 ± 3.04%, $p = 0.499$, $N = 3$ independent experiments). **j** *Bcl-xl*, *c-Flip*, *Bax*, and *Chop* mRNA levels in cultured WT and ATG5 KO neurons. $N = 5$ independent experiments. mRNA levels were normalized to the levels of housekeeping gene *Tbp* set to 100%. **k** ATG16L1 protein levels in lysates from cultured WT and ATG16L1 KO neurons. **l** Cleaved CASP3 levels in lysates from cultured WT and ATG16L1 KO neurons, starved for 16 h or left untreated. HEK293T cells treated with $H_2O_2$ were used as a positive control. **m**, **n** Histopathological analysis of Nissl-stained cortical sections of ATG16L1 deficient brains at 13-weeks reveals no morphological alterations and unchanged number of neurons (WT: 0.0021 ± 0.0001, KO: 0.0023 ± 0.0001, $p = 0.354$, $N = 3$ independent experiments). Scale bar, 200 μm. All graphs show mean ± SEM, statistical analysis was performed by unpaired two-tailed Student's *t*-test. n.s.-non-significant. Source data are provided as a Source Data file.

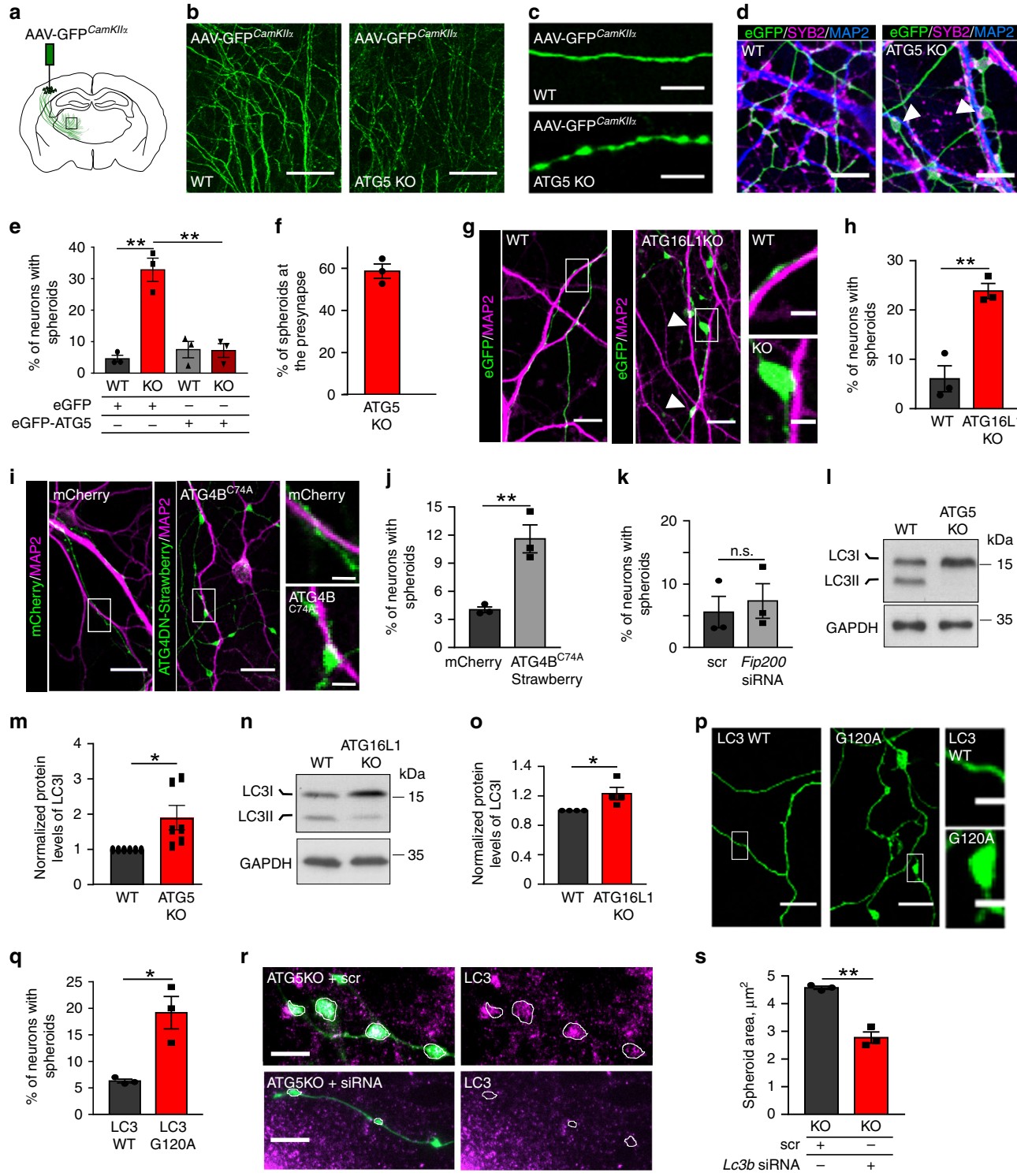

lethality (Supplementary Fig. 2a). To obtain further insights into neuronal function of ATG5 we analyzed the morphology of 13-week-old KO brains in detail. Although the anatomy of the cortex and the hippocampus was not altered by the ATG5 deletion (see Fig. 1), we detected a substantial loss of neurons in the thalamus (the posterior and ventral nuclei) (Supplementary Fig. 2b), an area, which is not targeted by the activity of Cre expression-driving *CamIIα* promoter (Supplementary Fig. 2c). Since thalamic atrophy occurs in a number of axonal dystrophy-associated diseases[25,26], we hypothesized that structural changes observed in ATG5 KO thalamus were caused by the degeneration of long-

range corticothalamic axons. To test this hypothesis, we labeled efferent projections of cortical deep layer neurons to the thalamus via stereotactic injection of an adeno-associated virus, expressing GFP under *CamKIIα* promoter (AAV-GFP*CamKIIα*, Fig. 2a). Indeed, corticothalamic axons of ATG5-deficient neurons revealed signs of severe axonal degeneration, characterized by en-passant swellings (Fig. 2b, c). Appearance of axonal swellings, termed axonal spheroids, was also reproduced in in-vitro ATG5 deficient cortico-hippocampal culture system (Fig. 2d, Supplementary Fig. 2d), where about 30% of neurons displayed signs of axonal injury (Fig. 2e, Supplementary Fig. 2e) mostly confined to

**Fig. 2 Loss LC3 lipidation machinery causes axonal neurodegeneration. a** Schematic illustration of stereotaxic delivery of AAV9-GFP$^{CamKIIa}$ into deep layers of primary motor cortex. **b, c** Loss of ATG5 causes en-passant swellings of long-range projection axons. Scale bar, (**b**) 200 μm, (**c**) 10 μm. **d** eGFP-transfected WT and ATG5 KO neurons immunostained for MAP2 and SYB2. Scale bar, 50 μm. **e** Percentage of WT and ATG5 KO neurons revealing axonal swellings when transfected either with eGFP (WT: 4.60 ± 1.09%, KO: 32.82 ± 3.70%) or with ATG5-eGFP (WT: 7.47 ± 2.60%, KO: 7.18 ± 2.20%). \*\*$p$WT$^{GFP}$ vs. KO$^{GFP}$ = 0.001, \*\*$p$WT$^{ATG5}$ vs. KO$^{ATG5}$ = 0.002. $N$ = 3 independent experiments. **f** Axonal swellings are mostly found at presynapses (58.62 ± 0.81%). **g, h** Percentage of eGFP-transfected WT and ATG16L1 KO neurons with axonal swellings (WT: 6.05 ± 2.14%, KO: 23.85 ± 1.24%), \*\*$p$ = 0.004, $N$ = 3 independent experiments. Scale bars: 10 μm, inserts 2 μm. **i, j** Percentage of mCherry-ATG4B$^{C74A}$ or pmStrawberry-ATG4B$^{C74A}$ (green)-transfected neurons revealing axonal swellings (mCherry: 4.03 ± 0.25%, pmStrawberry-Atg4BC74A 11.60 ± 1.22%). \*\*$p$ = 0.008. $N$ = 3 independent experiments. Scale bars: 10 μm, inserts 2 μm. **k** Percentage of axonal swellings in neurons with *Fip200* siRNA-mediated KD (scr:4.46 ± 2.40%, siRNA:5.78 ± 2.87%). $p$ = 0.659. $N$ = 4 independent experiments. **l–o** LC3I protein levels are significantly increased in ATG5 KO (1.91 ± 0.35, \*$p$ = 0.023, $N$ = 6 independent experiments) and ATG16L1 KO (1.23 ± 0.08, \*$p$ = 0.04, $N$ = 4 independent experiments) lysates, compared to WT controls set to 1. **p, q** Percentage of ptagRFP-LC3B (proLC3)- or ptagRFP-LC3B G120A-expressing neurons with axonal swellings (ptagRFP- LC3B: 6.29 ± 0.30%, ptagRFP-LC3B G120A: 19.16 ± 2.50%). \*$p$ = 0.014. $N$ = 3 independent experiments. Neurons were co-transfected with eGFP to visualize the axons. Scale bars: 4 μm, kymographs, 5 μm × 20 s. **r, s** Knockdown of LC3B in ATG5 KO neurons significantly decreases axonal spheroid area (KO$^{scr}$: 4.57 ± 0.06 μm$^2$, KO$^{siLC3b}$2.78 ± 0.20 μm$^2$). \*\*$p$ = 0.001. $N$ = 3 independent experiments. eGFP-expressing ATG5 KO neurons were treated either with scramble siRNA (scr) or *LC3b* siRNA and immunostained for LC3. Circles indicate en-passant axonal swellings. Scale bars: 5 μm. All graphs show mean ± SEM, statistical analysis was performed by unpaired two-tailed Student's $t$-test in (**h, j, k, q, s**), two-way ANOVA for multiple comparisons in (**e**) and by one-sample Student's $t$-test in (**m, o**). n.s.-non-significant. Total number of neurons in N experiments is shown in Supplementary Table 3. Source data are provided as a Source Data file.

---

presynaptic terminals (Fig. 2f). This phenotype was rescued by re-expression of ATG5-eGFP (Fig. 2e). Neither dendritic morphology, nor spine density were affected by the ATG5 deletion (Supplementary Fig. 2f, g). Axonal swellings were also evident in about 24% of cultured ATG16L1 KO neurons (Fig. 2g, h) and in neurons overexpressing an inactive C74A form of ATG4B[27], which inhibits the lipidation of LC3 (Fig. 2i, j). In contrast, deletion of ULK-interacting protein FAK-family kinase-interacting protein of 200 kDa (FIP200), a manipulation that prevents autophagy induction, not directly affecting the levels of cytoplasmic LC3[28], was not sufficient to cause the formation of axonal swellings (Fig. 2k, Supplementary Fig. 2h, i). These data indicate that LC3 lipidation machinery regulates axonal homeostasis and suggest that the increased abundance of cytoplasmic LC3 (LC3I) in the absence of ATG5 and/or ATG16L1 (Fig. 2l–o, see also Supplementary Fig. 1d for in-vivo levels) might be responsible for the appearance of swellings in axons. In agreement with this hypothesis, axonal degeneration was observed in control neurons overexpressing the lipidation-deficient mutant of LC3B (G120A), but was absent in neurons overexpressing the WT LC3 (Fig. 2p, q, Supplementary Fig. 2j), suggesting that under control conditions the autophagy lipidation machinery maintains the equilibrium between cytoplasmic and lipidated LC3 isoforms. In fact, shifting the balance towards cytoplasmic LC3 by treating the eGFP-LC3B transgenic neurons (Supplementary Fig. 2k) with cell membrane permeable LC3B peptide, which would titrate ATG4 in a dominant-negative manner and increase the availability of cytoplasmic LC3B, mimicked the axonal pathology phenotype (Supplementary Fig. 2l, m). Finally, downregulation of LC3B, using a smart pool of short interfering (si) RNAs directed against *Lc3b*, diminished axonal pathology in ATG5 KO neurons (Fig. 2r, s, Supplementary Fig. 2n, o).

**Degeneration of KO axons is protein inclusions-independent.** What is the mechanism by which loss of autophagy lipidation machinery causes axonal dystrophy? Accumulation of protein aggregates and/or aberrant pre-autophagosomal structures[29] is a classical hallmark of defective autophagy. We thus hypothesized that axonal degeneration is caused by defective clearance of ubiquitinated proteins and stalled pre-autophagosomal intermediates in axonal terminals. To test this, we first examined the levels of the autophagy receptor p62/SQSTM1 and ubiquitin in soma and axons of autophagy-deficient neurons by immunocytochemistry. While massive accumulation of cytoplasmic p62-positive inclusions was observed in the soma of ATG5 and ATG16L1 KO neurons

(Fig. 3a–e, Supplementary Fig. 3a, b), axons of autophagy-deficient neurons or neurons overexpressing the C74A mutant of ATG4B were devoid of ubiquitin and p62 inclusions (Fig. 3f–h, Supplementary Fig. 3c–f). Next, we analyzed pre-autophagosomal structures in KO neurons using antibodies recognizing early autophagosomal intermediates. In agreement with unaltered levels of FIP200, ATG13, WIPI2 in KO lysates (Supplementary Fig. 3g), we found that KO swellings did not accumulate early autophagosomal intermediates (Supplementary Fig. 3h–o). The levels of ATG9 and WIPI2 were even significantly lower in KO axons, likely due to impaired membrane trafficking from the soma (Supplementary Fig. 3i, o). These results are in line with previously published data, revealing the neurodegeneration in protein inclusion-free ATG7 KO brains[16] and strongly suggest that axonal pathology in neurons lacking LC3 lipid conjugation machinery is caused via protein inclusions-independent mechanism.

**KO axons accumulate components of trafficking machinery.** Next, we hypothesized that axonal swellings might represent distended synapses, formed either due to defective synaptic vesicles (SV) exocytosis or as a result of impaired vesicular trafficking. To test the first hypothesis, we employed the pH-sensitive fluorescent protein pHluorin fused to the Synaptobrevin2 (SYB2), which is widely used as a reporter of SV exocytosis[30]. Analysis of SYB2-pHluorin decay fluorescence upon stimulation with 200 action potentials at 50 Hz revealed that autophagy-deficient neurons are still capable of undergoing SV exocytosis (Supplementary Fig. 3p).

Ultrastructural studies indicate that axonal dystrophy is often accompanied by the accumulation of membranous organelles[31], including mitochondria and late endosomes[32,33]. In fact, we found that ATG5 KO presynaptic terminals examined by electron microscopy (EM) accumulated late endosomal organelles (Fig. 3i) and functional mitochondria (Supplementary Fig. 3q–s), a phenotype, which was confirmed by immunostaining with late endosomal marker RAB7 (Fig. 3j, k). Levels of RAB7 in the soma of ATG5 KO neurons were not altered (Supplementary Fig. 3t). Since RAB7 is known to play an essential role in the MT-based axonal transport[34], we hypothesized that axonal swellings in autophagy-deficient neurons might result from defective MT-based trafficking of intracellular cargo. To elucidate this hypothesis, we first examined the levels of MT-associated dynein activator Dynactin1 (DYNC1) in axonal spheroids of ATG5 KO neurons. Our data revealed that while in control condition DYNC1 showed a homogeneous cytosolic appearance along the

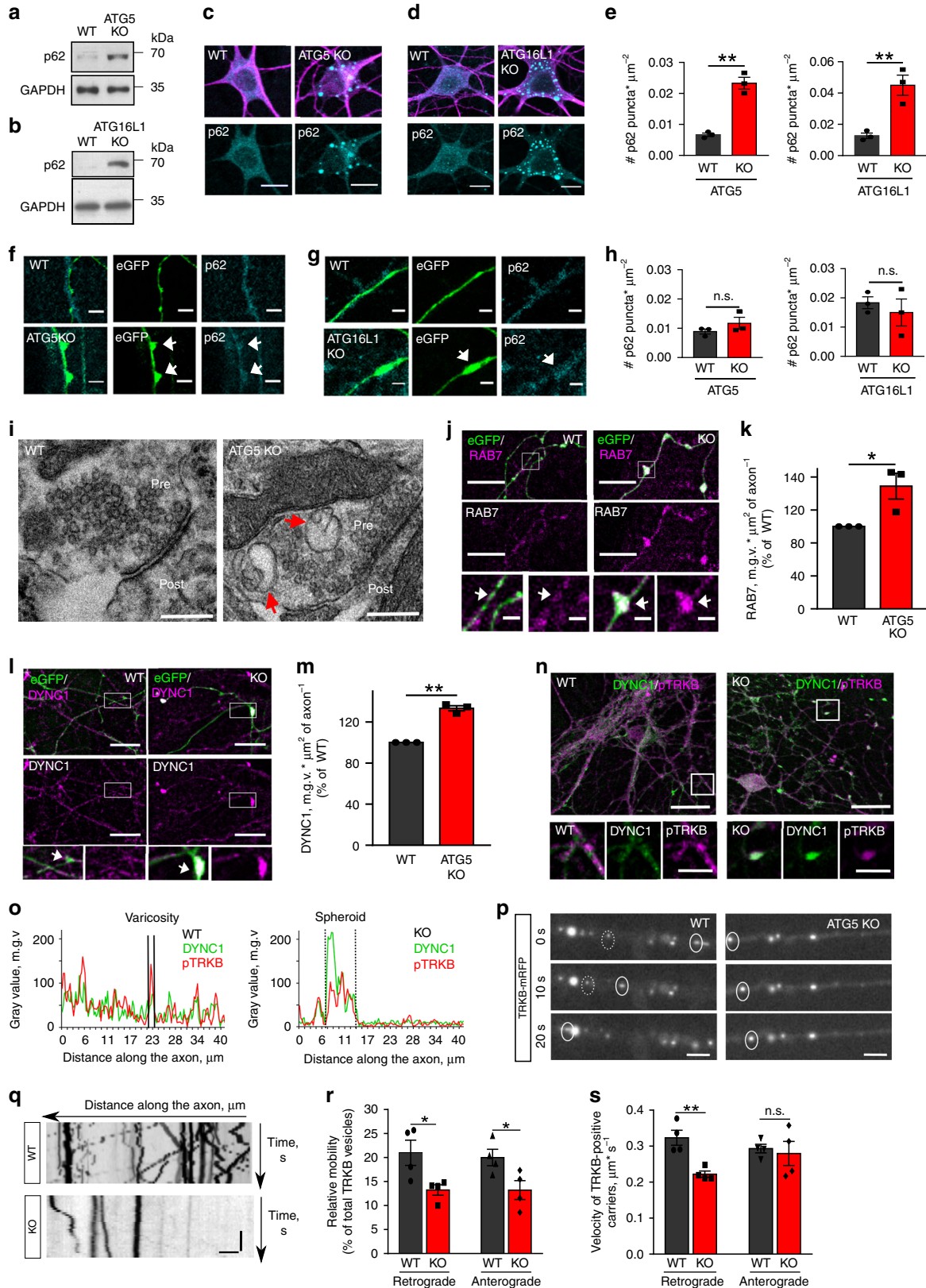

axons, ATG5 KO axons revealed large 5–10 μm spheroid-like accumulations of DYNC1 (Fig. 3l, m). Furthermore, in agreement with the presence of endosome-like membrane organelles at the KO synapse, we found that KO swellings revealed an increased colocalization of DYNC1 with activated tropomyosin-related kinase receptor B (TRKB) receptors (Fig. 3n, o, Supplementary

Fig. 3u), a known cargo of dynein motors in axons[35]. This phenotype was specific to MT-based axonal cargo, since the localization of presynapse-confined SV protein SYB2 was not altered by the ATG5 deletion (Supplementary Fig. 3v, w).

To obtain further insights into the ATG5 function in transport of axonal cargo we monitored the dynamics of fluorescently-tagged

**Fig. 3 Axonal swellings in autophagy-deficient neurons contain components of MT-based trafficking machinery. a, b** p62 levels in lysates from cultured WT, ATG5 KO (**a**) or ATG16L1 KO (**b**) neurons. **c–e** Analysis of p62-containing puncta in somata of ATG5 KO and ATG16L1 KO neurons, immunostained for MAP2 (magenta) (WT$^{ATG5}$: 0.007 ± 0.001, KO$^{ATG5}$: 0.023 ± 0.002, **$p$ = 0.001; WT$^{ATG16L1}$: 0.013 ± 0.002, KO$^{ATG16L1}$: 0.045 ± 0.007, **$p$ = 0.009), $N = 3$ independent experiments. Scale bars: 5 μm. **f–h** Analysis of p62-containing puncta in axons of ATG5 KO and ATG16L1 KO eGFP-transfected neurons (WT$^{ATG5}$: 0.009 ± 0.001, KO$^{ATG5}$: 0.012 ± 0.002, $p$ = 0.131; WT$^{ATG16L1}$: 0.018 ± 0.002, KO$^{ATG16L1}$: 0.015 ± 0.005, $p$ = 0.545), $N = 3$ independent experiments. Scale bars: 5 μm. **i** Electron micrographs of WT and ATG5 KO synapses. Endosomal intermediates are marked by red arrows. Scale bar: 200 nm. **j, k** RAB7 puncta density in WT and ATG5 KO axons (KO: 148.84 ± 15.63%). *$p$ = 0.044. $N = 3$ independent experiments. Fluorescent levels in the KO were normalized to the WT set to 100%. Scale bars (**j**): 10 μm, 2 μm inserts. **l, m** DYNC1 puncta density in WT and ATG5 KO axons (KO: 133.26 ± 2.53%). **$p$ = 0.003. $N = 3$ independent experiments. Fluorescent levels in the KO were normalized to the WT set to 100%. Scale bars (**l**): 10 μm, 2 μm inserts. **n, o** Confocal images (**n**) and fluorescent profiles (**o**) of WT and ATG5 KO neurons immunostained for DYNC1 and phosphorylated TRKB receptors (pTRKB). Scale bars: 30 μm, 10 μm inserts. **p, q** Time-lapse images and corresponding kymographs of TRKB-mRFP-expressing WT and ATG5 KO neurons. Scale bars: $x$: 5 μm, $y$: 10 s. **r** Loss of ATG5 significantly decreased the mobility of TRKB carriers compared to WT controls (WT$^{Retro}$: 20.99 ± 2.63%, KO$^{Retro}$: 13.23 ± 1.09%, WT$^{Antero}$: 20.00 ± 1.72%, KO$^{Antero}$: 13.22 ± 1.97%). *$p^{Retro}$ = 0.034, *$p^{Antero}$ = 0.041, $N = 4$ independent experiments. **s** Loss of ATG5 significantly decreased the retrograde TRKB velocity compared to WT controls (WT$^{Retro}$: 0.32 ± 0.02 μm * s$^{-1}$, KO$^{Retro}$: 0.22 ± 0.01 μm * s$^{-1}$, WT$^{Antero}$: 0.29 ± 0.01 μm * s$^{-1}$, KO$^{Antero}$: 0.28 ± 0.03 μm * s$^{-1}$). **$p^{Retro}$ = 0.004, $p^{Antero}$ = 0.721. $N = 4$ independent experiments. All graphs show mean ± SEM, statistical analysis was performed by unpaired two-tailed Student's $t$-test in (**e, h, r, s**) and one-sample Student's $t$-test in (**k, m**). n.s.-non-significant. Total number of neurons in $N$ experiments is shown in Supplementary Table 3. Source data are provided as a Source Data file. m.g.v-mean gray value.

TRKB receptor (TRKB-mRFP) by live imaging (Fig. 3p, q). We found that while in control neurons TRKB-mRFP puncta displayed bidirectional dynein-dependent movement (Supplementary Fig. 3x, y) with a velocity between 0.3–0.4 μm * s$^{-1}$, ATG5 deletion significantly diminished the mobile fraction of retrograde and anterograde TRKB carriers (Fig. 3r) and caused a significant reduction in the retrograde velocity of TRKB receptors (Fig. 3s). Of note, the TRKB velocity was faster in young neurons than that of mature neurons (Supplementary Fig. 3z). Similar results were obtained by analyzing the axonal transport of mitochondria in ATG5-deficient neurons (Supplementary Fig. 3a′, b′). Taken together, these results demonstrate that axonal pathology in neurons lacking the LC3 lipid conjugation machinery is accompanied by the distal accumulation of components of MT-based trafficking machinery.

**LC3 lipidation machinery regulates MT dynamics in neurons**. Our findings above raise the question of whether the LC3 lipidation machinery directly regulates tubulin dynamics. The fundamental difference between dendrites and axons is the extent of "MT dynamic instability". While dendrites contain equal amounts of stable and dynamic MTs, axonal MTs are mostly stable at the axonal shaft and labile at the en passant presynaptic terminals[36]. Given the fact that the pathology in autophagy deficient neurons is most evident at presynaptic axonal boutons, we next asked whether disruption of ATG5 alters the dynamic properties of axonal MTs. To this aim, we analyzed MT polymerization events in control and ATG5 KO axons, transfected with plus-end MT binding protein EB3-dtTomato (Fig. 4a, b), a reporter of MT polymerization events (comets)[37]. We found that while in control condition EB3-positive comets were evident along the axons, the overall number of EB3-positive comets was severely reduced under autophagy-deficient condition, a phenotype rescued after re-expression of ATG5 (Fig. 4c). Next, we probed the total amount of dynamic MTs in ATG5 KO neurons, by extracting the labile MTs and analyzing the levels of total α-tubulin, as well as detyrosinated (stable MTs) and tyrosinated (dynamic MTs) α-tubulin by Western blotting. As shown in Fig. 4d, e, the total amount of dynamic MT was significantly reduced under autophagy-deficient conditions.

A prediction from these data is that ATG5 KO neurons reveal generally higher levels of stable MTs, which in neurons are known to be posttranslationally modified to detyrosinated and Δ2 α-tubulin (lacking two C-terminal amino acids). As predicted the amount of Δ2 α-tubulin was significantly upregulated in neurons lacking ATG5 (Fig. 4f–h). Furthermore, ATG5 KO neurons

revealed higher levels of acetylated tubulin (Supplementary Fig. 4a, b) and were resistant to nocodazole treatment (Supplementary Fig. 4c, d), a drug which induces MT disassembly. Total levels of α-tubulin were altered neither in ATG5 nor in ATG16L1 KO neurons (Supplementary Fig. 4e–h). Finally, we separated polymerized MTs from soluble fraction of tubulin (monomers and dimers) in cultured ATG5 KO neurons and analyzed the MT fractions as above (Fig. 4i). We found that level of stable MTs was significantly increased in autophagy-deficient neurons, while the labile MTs, probed by an antibody against the tyrosinated tubulin, were reduced (Fig. 4j).

Taking into account the effect of other components of LC3 lipidation machinery on the formation of axonal swellings (see also Fig. 2q, r), we next analyzed MTs stability in ATG16L1 KO neurons. Similarly to ATG5 deficient neurons, ATG16L1 KO cells revealed higher levels of Δ2 α-tubulin (Fig. 4k–m), while the overall number of EB3-positive comets was significantly reduced (Fig. 4n–p). ATG16L1 deficient neurons were also resistant to nocodazole treatment (Supplementary Fig. 4i). EB3-positive comet density was additionally decreased in wild type neurons overexpressing LC3B G120A, but not in cells overexpressing LC3B wildtype (Fig. 4q, r) or lipidation-deficient GABARAP (Fig. 4s, t).

**Cytoplasmic LC3A/B associate with ELKS1 in the brain**. The data described thus far suggest a model according to which the levels of cytoplasmic LC3 regulated via autophagy lipidation machinery fine-tune the MT dynamics in neurons. Given that LC3 was originally identified as a MT-binding protein, which can both indirectly and directly associate with tubulin and MTs[38–40], we hypothesized that hyperstable MTs found in neurons under autophagy-deficient conditions are the result of increased association of LC3 with neuronal tubulin. To probe this hypothesis, we carried out co-immunoprecipitation experiments using soluble and polymerized MT fractions isolated from ATG5 WT and KO neurons. Although a small amount of LC3 was found to be co-immunoprecipitated with α-tubulin in the soluble MT fraction, we could not detect an upregulation of LC3 association with MTs in absence of ATG5 (Supplementary Fig. 5a, b). Thus, the MT stability phenotype under ATG5-defective conditions does not appear to result from increased LC3 association with neuronal tubulin.

To further dissect the molecular mechanism underlying the regulation of MT dynamics by cytoplasmic LC3 we performed mass spectrometry analysis of LC3 binding partners in the adult mouse brain (Fig. 5a, Supplementary Fig. 5c, see also data

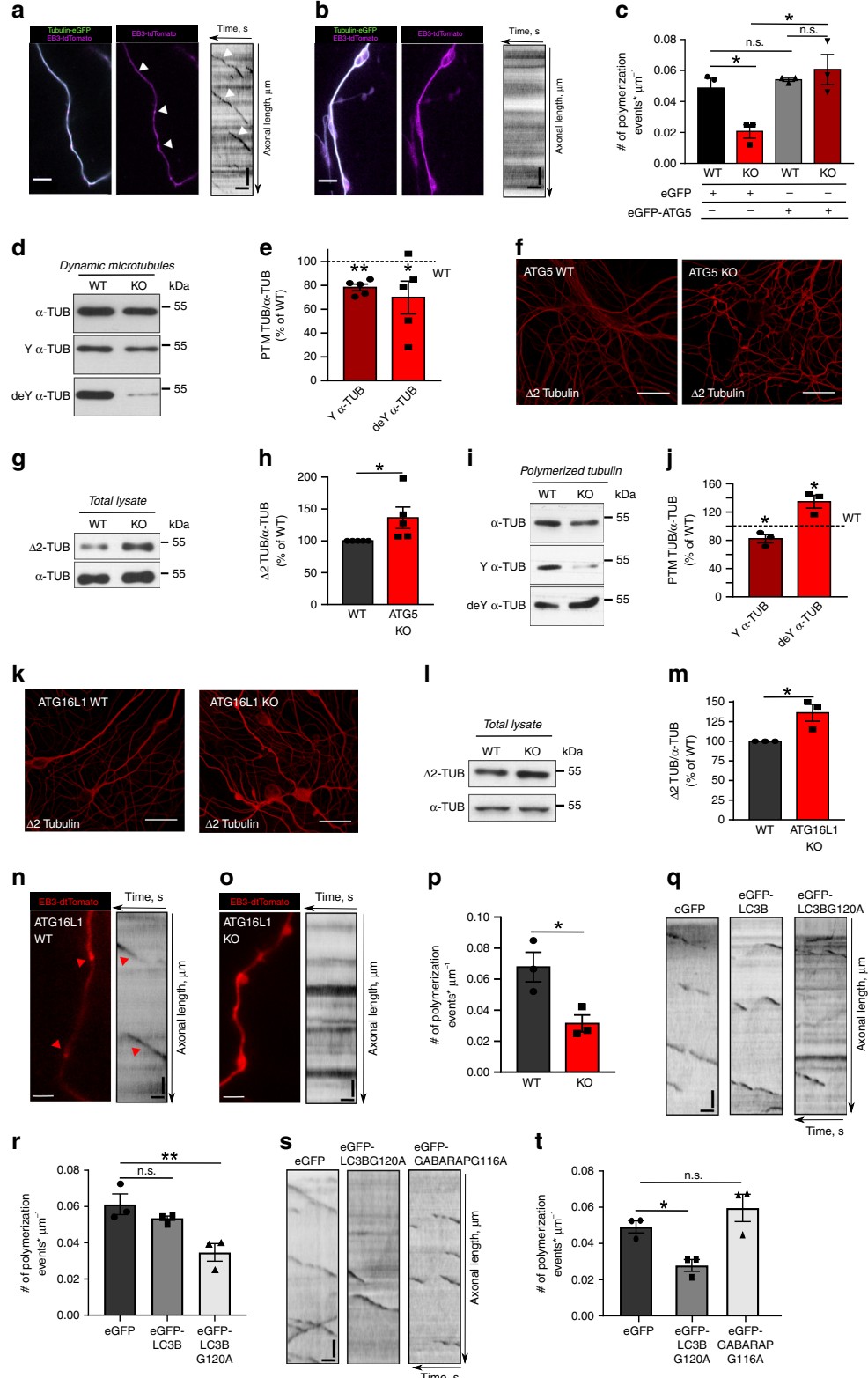

deposited in the ProteomeXchange, #PXD011279). In agreement with the data from cultured neurons, we failed to detect specific binding of neuronal LC3 to tubulin subunits in the brain (Supplementary Fig. 5e–k). Interestingly, we identified a protein called ELKS1 (glutamic acid/leucine/lysine/serine-rich protein, also known as RAB6IP2, CAST2 or ERC1[41]), as a putative interactor of LC3 in the brain (Fig. 5a, Supplementary Fig. 5d). ELKS1 is a scaffolding protein, which is ubiquitously expressed in

all tissues, but is most abundant in neurons, where it is enriched at the presynaptic active zone (AZ)[42] (Supplementary Fig. 6a,b). To determine the role of LC3 homologs in the regulation of ELKS1 function, we first analyzed their interactions using co-immunoprecipitation assays and colocalization studies. Ectopically expressed ELKS1, tagged with tdTomato was co-immunoprecipitated with eGFP tagged LC3A and LC3B, but not with GABARAP (Fig. 5b–d). We observed that both

**Fig. 4 LC3 lipid conjugation machinery regulates MT dynamics. a, b** Time-lapse images and corresponding kymographs of WT and ATG5 KO axons co-expressing Tubulin-eGFP and EB3-tdTomato. Arrows indicate EB3 comets in WT neurons. **c** EB3 comet density in WT and ATG5 KO axons, expressing either eGFP (WT$^{GFP}$: 0.05 ± 0.01, KO$^{GFP}$: 0.02 ± 0.00) or eGFP-ATG5 (WT$^{ATG5}$: 0.05 ± 0.00, KO$^{ATG5}$: 0.06 ± 0.01). *$p$WT$^{GFP}$ vs. KO$^{GFP}$ = 0.032, *$p$KO$^{GFP}$ vs. KO$^{ATG5}$ = 0.019. $N = 3$ independent experiments. **d, e** Levels of tyrosinated (Y) (78.28 ± 3.12%, **$p = 0.001$) and detyrosinated (deY) (69.85 ± 13.75%, *$p = 0.047$) α-Tubulin in dynamic ATG5 KO MTs. $N = 5$ independent experiments. **f–h** Δ2α-Tubulin levels in immunostained (**f**) and lysed (**g, h**) cultured WT and ATG5 KO neurons (KO: 136.13 ± 16.78%). *$p = 0.049$. $N = 5$ independent experiments. Scale bars, 50 μm. **i, j** Levels of tyrosinated (Y) (82.25 ± 5.75%, *$p = 0.045$) and detyrosinated (deY) (134.49.25 ± 9.09%, *$p = 0.032$) α-Tubulin in polymerized ATG5 KO MTs. $N = 3$ independent experiments. **k–m** Δ2α-Tubulin levels in in immunostained (**k**) and lysed (**l, m**) cultured WT and ATG16L1 KO neurons (KO: 136.18 ± 10.79%). *$p = 0.039$. $N = 3$ independent experiments. Scale bars: 50 μm. **n–p** Representative images, kymographs and comet density of EB3-tdTomato-expressing WT (0.07 ± 0.01) and ATG16L1 (0.03 ± 0.01) KO axons. *$p = 0.029$. $N = 3$ independent experiments. **q, r** Representative kymographs and comet density in EB3-tdTomato-expressing axons, transfected with either eGFP (0.061 ± 0.006), eGFP-LC3B (0.054 ± 0.001) or eGFP-LC3B G120A (0.035 ± 0.005). **$p^{eGFP}$ vs. eGFP-LC3B G120A = 0.009, $p^{eGFP}$ vs. eGFP-LC3B = 0.413. $N = 3$ independent experiments. **s, t** Representative kymographs and comet density in EB3-tdTomato-expressing axons, co-transfected either with eGFP (0.049 ± 0.003), eGFP-LC3B G120A (0.028 ± 0.003) or eGFP-GABARAP G116A (0.060 ± 0.008). *$p^{eGFP}$ vs. eGFP-LC3B G120A = 0.044, $p^{eGFP}$ vs. eGFP-GABARAP G116A = 0.321. $N = 3$ independent experiments. All graphs show mean ± SEM, statistical analysis was performed by unpaired two-tailed Student's $t$-test in (**p**), two-way ANOVA for multiple comparisons in (**c**), one-way ANOVA for multiple comparisons in (**r, t**) and one-sample Student's $t$-test in (**e, h, j, m**). n.s.-non-significant. In (**e, h, j, m**) KO protein levels were normalized to the WT set to 100%. In (**e, h, j, m**) samples arise from the same experiment and the blots were processed in parallel such that one loading control was used. Total number of neurons in $N$ experiments is shown in Supplementary Table 3. Source data are provided as a Source Data file. Scale bar for all kymographs $x$: 5 μm, $y$: 20 s.

unlipidated LC3A G120A and LC3B G120A bind ELKS1, with a stronger preference of LC3B G120A for ELKS1 (Fig. 5e, f), while the absence of binding was detected when the lipidation-deficient GABARAP was used as a bait (Fig. 5g). Reciprocally, eGFP-LC3A and eGFP-LC3B were co-immunoprecipitated with tdTomato-ELKS1 (Fig. 5h, i). Subsequent pull-down experiments with native proteins in vitro confirmed that the interaction between LC3B and ELKS1 is direct (Fig. 5j); while co-immunoprecipitation studies and immunofluorescence analysis in mouse neurons indicated that endogenous LC3 and ELKS can associate in the brain in-vivo (Fig. 5k–n). Interestingly, only cytoplasmic LC3 was found in the complex with ELKS1 in neuronal cells (Fig. 5l), where they both localized to *en passant* presynaptic terminals (Fig. 5m).

**CLASP2-dependent regulation of MTs stability via LC3/ELKS1.** Taken into account the ELKS1 association with LC3 in-vitro and in-vivo, we next analyzed ELKS1 levels in neurons lacking the LC3 lipid conjugation machinery. We found that slightly more ELKS1 was co-immunoprecipitated with LC3 in ATG5 KO neurons (Fig. 6a). Furthermore, impairment of LC3 lipid conjugation was generally associated with significantly higher levels of ELKS1 in cultured neurons (Fig. 6b, c, Supplementary Fig. 6c) and in brains of ATG5 KO mice (Supplementary Fig. 6d, e), where it was enriched at the presynaptic boutons (Fig. 6d–f). Interestingly, the degradation of ELKS1 was autophagy-independent per se (Fig. 6g), but was found to require the proteasome (Supplementary Fig. 6f, g), similarly to its *Drosophila* homolog Bruchpilot[43]. Furthermore, in cells over-expressing the LC3B G120A mutant and treated with the proteasome inhibitor MG132 the amount of ELKS1 was significantly higher than that in LC3B wildtype-expressing cells (Supplementary Fig. 6h, i). LC3B G120A also decreased ELKS1 protein turnover (examined upon inhibition of protein synthesis using cyclohexamide) (Supplementary Fig. 6k, m), while deletion of LC3B in ATG5 KO neurons via siRNAs smart pool-mediated gene KD normalized ELKS1 protein levels in axons (Fig. 6h–j).

An attractive scenario suggests that the autophagy-independent regulation of ELKS1 protein levels by LC3 lipid conjugation machinery is responsible for the MT stability phenotype observed earlier. In agreement with this hypothesis, deletion of ELKS1 in autophagy-deficient neurons was sufficient to rescue impaired MTs dynamics, monitored by overexpression of EB3-tdTomato in control and ATG5 KO neurons (Fig. 6k, Supplementary

Fig. 6n–q). These results taken together with the fact that the KD of both LC3A (Supplementary Fig. 6r) and LC3B (see Fig. 2s) decreases the area of axonal swellings in ATG5 KO neurons, indicate that LC3A/B lipid conjugation machinery non-canonically regulates MT dynamics via an ELKS1-dependent mechanism.

In non-neuronal cells ELKS regulates MT dynamics via its interaction with the MT-associated proteins CLASPs (Cytoplasmic linker associated proteins)[44]. Since endogenous ELKS1 and brain-enriched CLASP2[45] associate in the brain in-vivo (Supplementary Fig. 6s), we next speculated that axonal ELKS1 regulates MT dynamics via stabilizing the levels of its binding partner CLASP2. Consistent with our hypothesis, we found that the levels of CLASP2, tested by specific antibody (Supplementary Fig. 6t–w), were significantly upregulated in brains of autophagy deficient mice (Fig. 6n, o) and that CLASP2 was more abundantly associated with ATG5 KO MTs resolved by superresolution STED microscopy (Fig. 6p, Supplementary Fig. 6x). Although CLASP2 was not degraded by the proteasome per se, we observed that its protein levels were also stabilized by LC3B G120A in a proteasome-dependent manner (Supplementary Fig. 6j, l), while the siRNA-mediated downregulation of ELKS1 was sufficient to reduce the CLASP2 levels in ATG5 KO axons (Fig. 6q, Supplementary Fig. 6y). Finally, by performing an EB3 comet assay in CLASP2 overexpressing control neurons, we found that CLASP2 abundance directly decreases MT catastrophe frequency and increased the length of MT growth events (Fig. 6r–u), suggesting that CLASP2 protects MTs against depolymerization, by reducing the number of dynamic plus ends. CLASP2 was not regulating the polymerization rate of the remaining plus-ends, since the mean velocity of EB3 comets was not significantly altered by CLASP2 overexpression (Fig. 6v). These data, taken together with the fact that several MT plus-end tracking proteins, required for retrograde trafficking, localize to the ends of tyrosinated (dynamic), but not detyrosinated (stable) MTs[46], suggest that increased stabilization of MTs by CLASP2 without the ATG5 might impair the retrograde cargo trafficking in axons.

**Impaired BDNF/TRKB signaling in ATG5 KO brains.** Decreased MT dynamics in ATG5 KO neurons might have severe consequences for cargo delivery to the cell body and affect neuronal plasticity. A crucial pathway that promotes neuronal plasticity is the brain-derived neurotrophic factor (BDNF) signaling pathway. BDNF initiates signaling by binding to its receptor

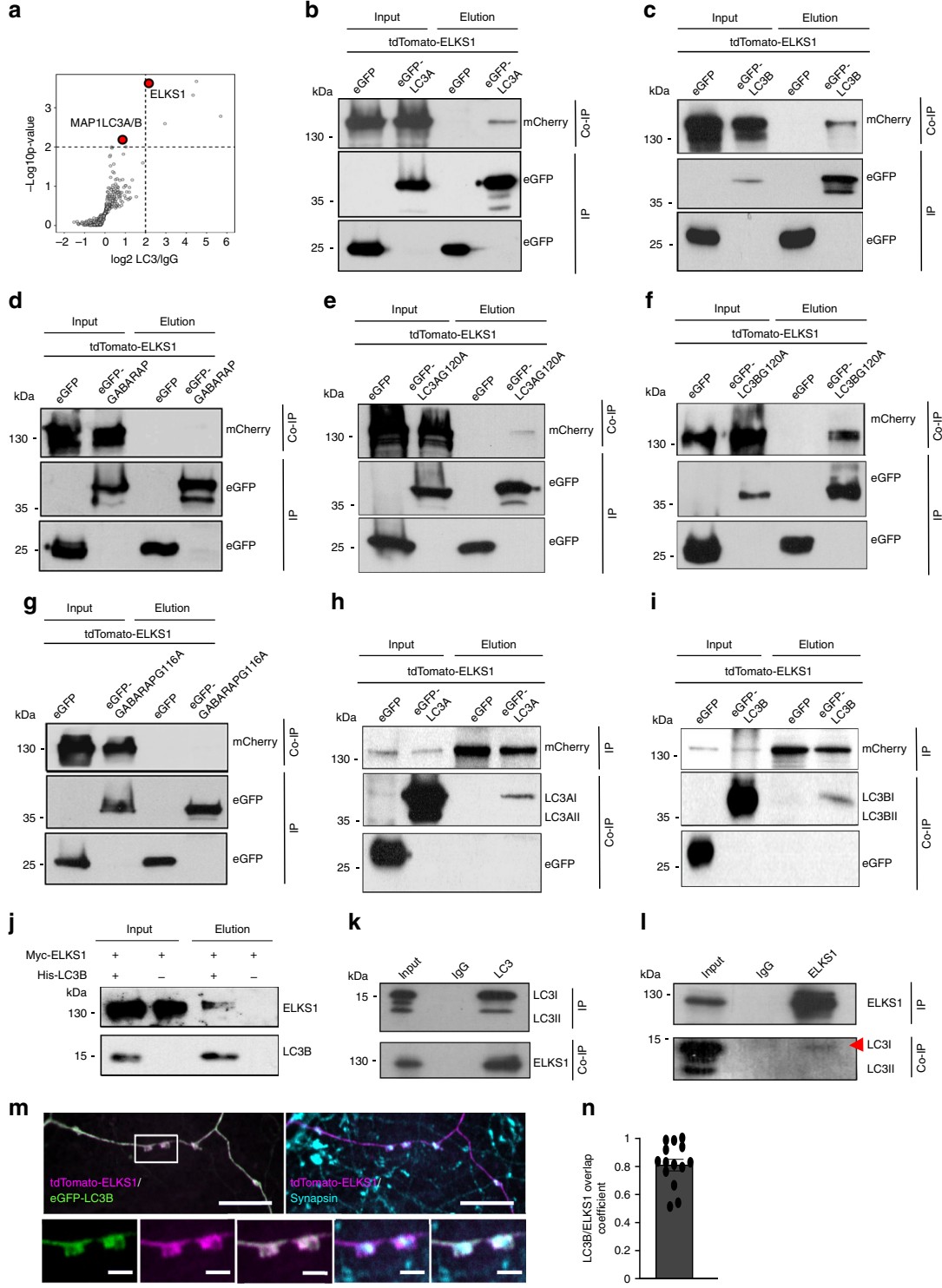

TRKB in distal neurites[47] and the retrograde transport of BDNF-activated TRKB receptors is required for the autocrine regulation of nuclear *Bdnf* expression[48]. Given the fact that retrograde transport of TRKB receptors is impaired in ATG5 KO neurons (see Fig. 3) one would expect the BDNF/TRKB signaling to be reduced in these cells. As predicted, we observed that levels of BDNF-activated TRKB (p-TRKBY816), as well as the activation of its main downstream targets ERK1/2 and AKT were significantly decreased in ATG5-deficient neurons (Fig. 7a, b). In agreement with autocrine regulation of *Bdnf* gene expression, the levels of *Bdnf* mRNA, but not the mRNA levels of *TrkB* were

significantly reduced in the absence of ATG5 (Fig. 7c, d). This phenotype was also accompanied by a severe reduction in mTOR activity (Fig. 7e, f), likely as a result of diminished BDNF signaling.

Our previous data indicate that conditional loss of ATG5 results in defective neuronal arborization[35], a phenotype which might be a consequence of a failure in BDNF/TRKB signaling. Thus, we boosted the BDNF signaling through exogenous sustained application of BDNF to cultured WT and ATG5 KO neurons. Interestingly, although axonal dystrophy persisted under conditions of exogenous BDNF supply (Supplementary Fig. 7a), it

**Fig. 5 LC3 directly associates with ELKS1 in-vitro and in-vivo. a** Mass spectrometry analysis of LC3 interaction partners in the brain. Volcano plot illustrates the significantly abundant proteins. The $-\log_{10}$ is plotted against the $\log_2$ (fold change: LC3/control). The non-axial vertical lines denote ±2-fold change while the non-axial horizontal line denotes $p = 0.05$, which is the significance threshold. **b–d** Ectopically expressed ELKS1, tagged with tdTomato is co-immunoprecipitated (Co-IP) with eGFP-tagged LC3A (**b**) and LC3B (**c**), but not with GABARAP (**d**). HEK cell lysates were directly analyzed (Input) or subjected to GFP immunoprecipitation (Elution) and further immunoblotted against mCherry (Co-IP). Input, 1% of the total lysate. **e–g** Ectopically expressed ELKS1, tagged with tdTomato is co-immunoprecipitated (Co-IP) with eGFP-tagged LC3A G120A (**e**) and LC3B G120A (**f**) but not with GABARAP G116A (**g**). HEK cell lysates were directly analyzed (Input) or subjected to GFP immunoprecipitation (Elution) and further immunoblotted against mCherry (Co-IP). Input, 1% of the total lysate. **h, i** Ectopically expressed LC3A and LC3B, tagged with eGFP are co-immunoprecipitated with tdTomato-ELKS1. HEK cell lysates were directly analyzed (Input) or subjected to ELKS1 immunoprecipitation (Elution) and further immunoblotted against GFP (Co-IP). Input, 1% of the total lysate. **j** Purified recombinant His$_6$-LC3B, detected by immunoblotting with LC3B antibody, directly binds Myc-ELKS1. Input, 12.5% of the total recombinant LC3B and Myc-ELKS1. **k** Co-immunoprecipitation of endogenous ELKS1 with LC3B form mouse brain lysate. Input, 4% of the total lysate was added to the assay. **l** Co-immunoprecipitation of endogenous LC3B with ELKS1 form mouse brain lysate. Input, 4% of lysate. **m, n** Close colocalization of tdTomato-ELKS1 and eGFP-LC3B at en passant presynaptic boutons marked by synapsin immunofluorescence (blue). Colocalization is analyzed by Mander´s overlap coefficient (80.42 ± 7.22%). Scale bars: 5 µm (upper panel), 1 µm (lower panels). $N = 2$ independent experiments. All graphs show mean ± SEM. Co-IP experiments in HEK cells represent examples from $N = 3$ independent experiments. Source data are provided as a Source Data file.

was sufficient to prevent the loss of dendritic complexity and induce mTOR activation in autophagy-deficient neurons (Fig. 7g–j, Supplementary Fig. 7b–d). Under similar conditions, nerve growth factor (NGF) application failed to rescue the mTOR signaling in ATG5 KO neurons (Supplementary Fig. 7e).

Finally, since dynamic MTs are crucial for synapse remodeling during long-term memory formation[11], a phenotype, which requires the BDNF/ERK1/2-dependent activation of immediate early genes, such as c-Fos[49], we hypothesized that impaired MT dynamics in combination with decreased BDNF/TRKB signaling will have severe consequences for memory acquisition in ATG5 deficient mice. To test this, we analyzed the c-Fos gene activation in the hippocampus of control and autophagy-deficient mice after acquiring the memory on a novel object in the open field arena (Novel Object Recognition test, NOR). We examined c-FOS protein levels 90 min after testing session performed 24 h after learning (Fig. 7k). The number of c-FOS positive neurons in the dentate gyrus was significantly decreased in mice conditionally lacking ATG5 (Fig. 7l), indicating that the ablation of the LC3 conjugation machinery in excitatory neurons arrests the NOR-dependent neuronal activity in the hippocampus.

## Discussion

Our data unravel an unexpected function of autophagy ubiquitin-like conjugation machinery in neurons. We show that ATG5 and ATG16L1 proteins, involved in LC3 lipid conjugation, are dispensable for survival of forebrain excitatory neurons. Instead, mammalian neurons capitalize on the autophagy lipidation machinery to regulate the stability of axonal MTs and maintain the branching complexity via modulating the transport of axonal cargo, including BDNF-activated TRKB receptors and mitochondria.

In neurons, stable MTs are known to provide tracks for cargo transport over long distances, while labile MTs are important for rapid reorganization of the cytoskeleton during synaptic plasticity, a function which can be especially crucial at highly dynamic en passant boutons[50], representing the majority of forebrain synapses[51]. Here we report that axonal pathology in autophagy-deficient neurons occurs mostly at cortical en passant–like synapses. Our data are consistent with the model according to which modest stabilization of MTs in the absence of the autophagy lipidation machinery prevents local remodeling of cytoskeleton at presynapses, a phenotype causing axonal swellings of long-distance cortical axons. Together with the observation that the reduction of dynamic MTs diminishes the transport of axonal cargo[52], our results predict that decrease in MT dynamics under autophagy-deficient condition impairs retrograde axonal

signaling required for synaptic plasticity. Indeed recent data have shown that MT-stabilizing drug Paclitaxel, which is used to treat a wide spectrum of tumors, including breast and ovarian cancer, causes dysregulation of BDNF signaling[53] accompanied by memory dysfunctions in mice[54] and humans[55], akin to the loss of ATG5 reported here.

In contrary to previously described direct association of LC3 with MTs[39], we find that regulation of MT stability in neurons is promoted by a protein complex comprising the AZ protein ELKS1 and its brain-enriched MT-binding partner CLASP2. CLASP2 belongs to the family of plus-end tracking proteins, which specifically accumulate at the growing MT plus-ends to mediate the MT attachment to cortical regions[56]. This function CLASPs requires their interaction with ELKS1 (also known as ERC1 or CAST2)[44], a protein, which in neurons constitutes a part of the cytomatrix of the AZ and is enriched at presynaptic terminals[42]. Several lines of evidence suggest that the regulation of neuronal MT dynamics requires the interaction of cytoplasmic LC3 with ELKS1. First, we demonstrate that neuronal LC3 associates with ELKS1 both in-vitro (Fig. 5j) and in-vivo (Fig. 5k, l). Second, upregulation of cytoplasmic LC3 levels stabilizes ELKS1 protein levels via inhibition of its proteasomal degradation (Fig. 6b–f, Supplementary Fig. 6f–m), while LC3 knockdown downregulates ELKS1 levels in ATG5 KO background (Fig. 6h–j). Finally, we demonstrate that knockdown of ELKS in autophagy-deficient neurons is sufficient to restore the impaired MT dynamics and diminish axonal pathology (Fig. 6k–m). The interaction of ELKS1 with LC3 is likely of special importance at cortical en-passant synaptic boutons, which maintain the neurotransmitter release with a frequency of 100 Hz[57]. Such intense activity places presynaptic proteins at a high risk of damage, a phenotype that requires effective retrograde transport of damaged and misfolded proteins to the cell soma for degradation[58]. Thus, we suggest a model according to which autophagy lipidation machinery functions at presynaptic terminals to regulate axonal retrograde transport by temporally adjusting the dynamics of MTs polymerization rates to intense cycles of exo/endocytosis of synaptic vesicles[30]. It is possible, if not likely, that the metabolic state of the presynapse will provide the precise temporal resolution for control of MTs-based transport in neurons. For instance, AMPK-dependent induction of autophagy[59] would decrease the amount of cytoplasmic LC3, thereby increasing the amount of dynamic MTs ends, which will facilitate the MT-based transport of axonal cargo.

Previously, indirect association between MTs and LC3 has been shown to require the MT-associated proteins MAP1A and MAP1B[38] or their homolog MAP1S[60]. Taken into account the fact that the expression of MAP1B is restricted to neonatal brain[61],

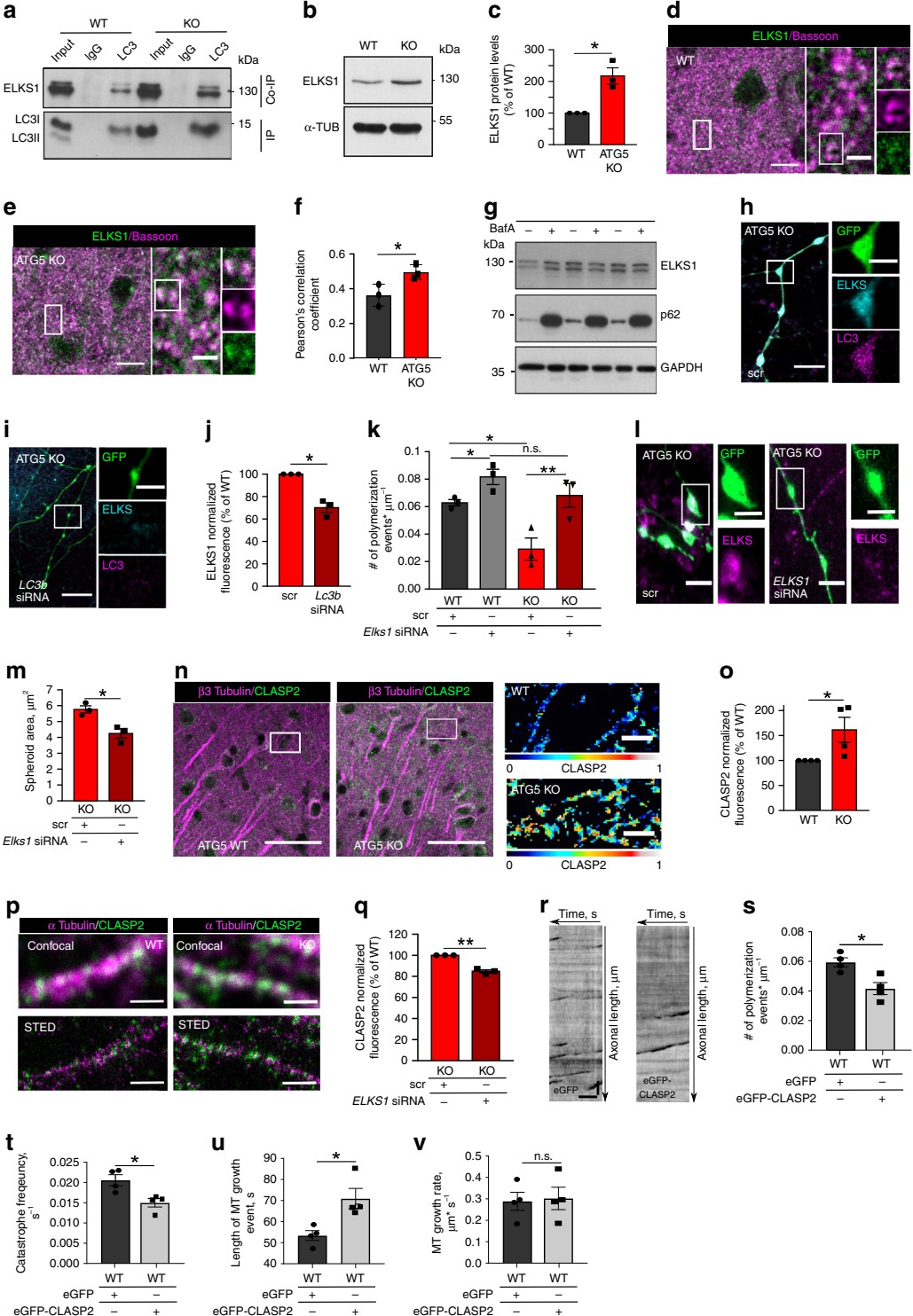

while MAP1A and MAP1S are predominantly localized to dendrites[62], our results provide a mechanism of how LC3 might regulate axonal MTs stability in the adult mammalian brain.

Neuronal-confined KO of several ATG genes causes neurodegeneration[63]. Despite of these findings and more than 50 years of autophagy research, the precise physiological function of ATG proteins in promoting neuronal survival remained elusive. Our data reveal a non-canonical role for autophagy lipidation machinery in regulation of MT stability in the axon and raise an intriguing possibility that neurodegeneration observed under autophagy-deficient conditions might at least partially arise from

altered MT dynamics. Further studies will be needed to test this hypothesis. However, based on our results we suggest that pharmacological activation of autophagy in the brain may not only promote the degradation of bulk cytoplasmic material, but also impair the axonal integrity via altering the MT stability.

## Methods

**Animals.** C57/BL/6 mice were housed in polycarbonate cages at standard 12/12 day-night cycles, and water and food were provided ad libitum. All animal experiments were approved by the ethics committee of LANUV Cologne and were conducted according to the committee's guidelines (2015.A578, 2016.A041, 2016.

**Fig. 6 LC3 regulates MTs stability via ELKS1/CLASP2-dependent mechanism. a** ELKS1 and LC3B Co-IP in WT and ATG5 KO lysates. Input, 10% of lysate. **b**, **c** ELKS1 levels in ATG5 WT and KO lysates. *$p = 0.022$. **d–f** ELKS1 colocalization with Bassoon in WT and ATG5 KO cortex, analyzed by Pearson's correlation coefficient. *$p = 0.04$. Scale bar, 10 μm, 2 μm inserts. **g** ELKS1 levels in NSC34 cells treated with 67 nm BafilomycinA for 16 h. **h–j** Decreased ELKS1 levels in eGFP-transfected ATG5 KO neurons, treated with *LC3b* siRNA, *$p = 0.010$. Scale bars, 10 μm, 4 μm inserts. **k** EB3 comet density in WT and ATG5 KO axons, transfected either with scr or *Elks1* siRNA. *$p\text{WT}^{scr}$ vs. $\text{KO}^{scr} = 0.025$, **$p\text{KO}^{scr}$ vs. $\text{KO}^{siElks1} = 0.004$, *$p\text{WT}^{scr}$ vs. $\text{WT}^{siElks1} = 0.041$. **l**, **m** Spheroid area is decreased in eGFP-transfected ATG5 KO axons treated with *Elks1* siRNA, compared to scr siRNA. *$p = 0.018$. Scale bars, 5 μm, 3 μm inserts. **n**, **o** Increased CLASP2 levels in ATG5 KO cortex. *$p = 0.046$. In (**n**), CLASP2 fluorescence was false color-coded with warm colors representing high intensities. Scale bars, 50 μm, 10 μm inserts. **p** STED imaging of CLASP2 localization on WT and ATG5 KO MTs. Scale bars, 1 μm. **q** CLASP2 levels in ATG5 KO axons treated either with scr or *Elks1* siRNA. **$p = 0.004$. **r–v** EB3 comet density (eGFP: $0.052 \pm 0.00$; eGFP-CLASP2: $0.0417 \pm 0.004$, *$p = 0.012$), MT catastrophe frequency (eGFP: $0.021 \pm 0.001\,\text{s}^{-1}$, eGFP-CLASP2: $0.015 \pm 0.001\,\text{s}^{-1}$, *$p = 0.017$), MT growth events length (eGFP: $53.39 \pm 2.35$ s, eGFP-CLASP2: $70.93 \pm 4.85$ s, *$p = 0.017$) and MT growth rate (eGFP: $0.289 \pm 0.042\,\mu\text{m}^{*}\text{s}^{-1}$, eGFP-CLASP2: $0.322 \pm 0.055\,\mu\text{m}^{*}\text{s}^{-1}$, $p = 0.654$) in neurons, transfected either with eGFP or eGFP-CLASP2. Scale bar, x: 5 μm, y: 20 s. Data in **b**, **c**, **f**, **j**, **k**, **m**, **q** are from $N = 3$ independent experiments. Data in **o**, **s–v** are from $N = 4$ independent experiments. All graphs show mean ± SEM, statistical analysis was performed by unpaired two-tailed Student's *t*-test in (**f**, **m**, **s–v**), two-way ANOVA for multiple comparisons in (**k**) and one-sample Student's *t*-test in (**c**, **j**, **o**, **q**). In (**c**, **o**) KO was normalized to WT, in (**j**, **q**) $\text{KO}^{si}$ was normalized to $\text{KO}^{scr}$. n.s.-non-significant. Total number of neurons in $N$ experiments is shown in Supplementary Table 3. Source data are provided as a Source Data file.

A451). Tamoxifen-inducible ATG5 KO mice (ATG5$^{flox/flox}$: B6.Cg-Tg(CAG-cre/Esr1*)5Amc/J) were described previously[35]. Forebrain confined ATG5 KO mice were created by crossing of *Atg5*$^{flox/flox}$ mice with the CamKIIα-Cre line[22]. *Atg16l1*$^{flox/flox}$ mice[64] were received from Prof. Philip Rosenstiel (University Hospital Kiel, Germany). Forebrain confined ATG16L1 KO mice were created by crossing of *Atg16l1*$^{flox/flox}$ mice with the CamKIIα-Cre line[22]. Tamoxifen-inducible ATG16L1 KO mice were created by crossing *Atg16l1*$^{lox/lox}$ mice with CAG-cre/Esr1 line (The Jackson Laboratory). Ai9(RCL-tdT) line was received from Dr. Matteo Bergami (CECAD, Cologne, Germany). GFP-LC3 mice (Riken) were described previously[65].

**Preparation of neuronal cultures and transfections**. Neuronal cell culture. Cortical and hippocampal neurons were isolated from postnatal mice at p1–5 as previously described[66]. In brief, neuronal cultures were prepared by surgically removing the cortices and the hippocampi from postnatal mice at p1-5, followed by trypsin digestion to dissociate individual neurons. Neurons were grown in a MEM (Gibco) medium, containing 0.5% Glucose (Sigma), 0.02% NaHCO3 (Roth) and 0.01% Transferrin (Merck), which was further supplemented with 5% FBS (Sigma), 0.25% L-GlutMax (100×, Gibco), 2% B27 (50×, Gibco), 1% Pen/Strep (10000 Units/ml Penicillin, 10,000 µg/ml Streptomycin, Gibco). After two days in-vitro 2 mM AraC was added to the culture medium to limit glial proliferation. To initiate homologous recombination, primary neurons isolated from floxed animals expressing a tamoxifen-inducible Cre recombinase were treated with 0.4 µM (Z)-4-hydroxytamoxifen (Sigma) immediately after plating. After 24 and 48 h, cells were treated with 0.2 µM and 0.4 µM of tamoxifen, respectively, during medium renewal. Ethanol was added to control neurons (WT) in an amount equal to the tamoxifen.

Plasmid and siRNA transfections. Neurons were transfected at DIV 7–9 days using an optimized calcium phosphate protocol with Calcium Phosphate transfection kit (Promega)[66]. For this 6 mg plasmid DNA, 250 mM CaCl2 and water (for each well of a 6-well plate) were mixed with equal volume of 2× HEPES buffered saline (100 ml) and incubated for 20 min allowing for precipitate formation, while neurons were starved in NBA medium for the same time at 37 °C, 5% CO2. Precipitates were added to neurons and incubated at 37 °C, 5% CO2 for 30 min. Finally, neurons were washed twice with HBSS medium and transferred back into their conditioned medium. In case of siRNAs smart pool knockdown experiments, eGFP plasmid was always co-transfected with 100 nM siRNAs (see Supplementary Table 2) on DIV-7-9 to identify the morphology of the axons in transfected cells, and neurons were always analyzed 5 days posttransfection.

Treatment with BDNF and ß-NGF. Neurons were supplied with 50 ng/mL of recombinant BDNF or ß-NGF (PAN Biotech) (the same amount of water was used as a control) at DIV 2, 6, 10, and 14. Neurons were fixed or harvested at DIV 16 for ICC or WB, respectively.

**Stereotaxic injection of AAV9-CamKIIα-eGFP**. Stereotactic injections were performed on 12–13-week-old *Atg5*$^{flox/flox}$:*CamKIIα*-Cre WT and KO mice. Mice were anesthetized with a mixture of Ketamin (100 mg/kg)/Xylazin (20 mg/kg)/Acepromazine (3 mg/kg) and mounted in a Kopf stereotactic frame. For the injection, a small hole was made in the skull of the mouse and a 1 µl Hamilton syringe filled with 300 nl of AAV9-CamKIIα-eGFP was lowered into the M1 area of primary motor cortex, using following coordinates: AP-1.25, DL-1.8, Depth-0.52. A volume of 300 nl AAV was delivered during 5 min, with a 5-min delay after the penetration, and waiting another 15 min before withdrawing of the syringe. The animal was given a dose of carprofen to reduce postsurgical pain (s.c, 5 mg/kg) before the end of the surgery. Suturing the skin over the wound completed the surgery and the animal was then allowed to recover. Animals were sacrificed by transcardial perfusion 10–14 days after surgery, and the eGFP expression was

analyzed by confocal microscopy (see Immunohistochemical analysis of brain sections).

**Plasmids**. Rat *LC3B* sequence (GeneID: 64862) was cloned in ptagRFP-C vector (Gift from Dr Michael Kreutz) using Bgl II and EcoRI restriction enzymes. For generating the LC3BG120A, Glycine at 120 was mutated to alanine, and a stop codon was introduced after the alanine 120 by side-directed mutagenesis. All plasmids used in the study are listed in the Supplementary Table2. Mouse *LC3A* sequence (Gene ID: 66734) was cloned into pEGFP-C vector (a gift from Dr Michael Kreutz) using EcoRI and BamHI restriction enzymes. For generating the LC3AG120A, Glycine at 120 was mutated to alanine. Mouse *GABARAP* sequence (GeneID: 56486) was cloned from cDNA derived from the cortex, using Hind III and BamHI restriction enzymes. GABARAPG116A construct was generated by mutating glycine at position 116 to alanine.

**Immunocytochemistry and analysis of cultured neurons**. Neurons were fixed at DIV 16–18 in 4% paraformaldehyde (PFA, Merck) in phosphate-buffered saline (PBS) for 15 min room temperature (RT), washed and blocked with PBS containing 5% normal goat serum (NGS, Thermo Scientific) and 0.3% Saponin (Serva) for 1 h. Neurons were incubated with primary antibodies (see Supplementary Table 1) for 1 h in PBS containing 10% NGS (Thermo Scientific) and 0.3% Triton X-100 (Roth). Coverslips were rinsed three times with PBS (10 min each) and incubated with corresponding secondary antibodies (see Supplementary Table 1) for 30 min (diluted 1:500 in PBS containing 0.3% TritonX and 10% NGS). Subsequently, coverslips were washed three times in PBS and mounted in Immu-Mount (Thermo Scientific)[35]. Fixed neurons were imaged using either Zeiss Axiovert 200 M microscope equipped with 40x/1.4 oil DIC objective and the Micro-Manager software (Micro-Manager1.4, USA) or with Leica SP8 confocal microscope (Leica Microsystems) equipped with a ×63/1.32 oil DIC objective and a pulsed excitation white light laser (WLL; ~80-ps pulse width, 80-MHz repetition rate; NKT Photonics). For quantitative analysis of fluorescent protein levels (or fluorescent puncta), the area of the neuron, cell body, axon or spheroid (depending the experiment) was manually selected using ImageJ selection tools (ROI) and the mean gray value was quantified within the ROI after the background subtraction. Spheroid was defined as axonal swelling larger than 2 µm in diameter. Fluorescent puncta were determined by applying the autothreshold "minimum" algorithm implemented in ImageJ and analyzed using the "Analyze particles" ImageJ module to determine the number of fluorescent puncta per 1 µm$^2$. For quantifying the number of neurons containing axonal swellings, the coverslips were imaged by using EVOS FL Auto 2 (Invitrogen, USA). For quantifying the number of spheroids per axon, the number and the diameter of spheroids from WT and KO neurons were analyzed for the last 300 µm of the axon by the ImageJ. The number of dendritic spines was quantified per 10 µm of primary dendrite per neuron.

**Immunostaining of α-tubulin in cultured neurons**. Neurons were treated on DIV 16 either with 0.2 µg/ml Nocodazole (Sigma) or with 0.2 µg/ml DMSO for 1 h. Afterwards, they were rinsed with warm PHEM buffer (60 mM PIPES, 25 mM HEPES, 10 mM EGTA, 4 mM MgCl2), followed by an incubation step in PHEM buffer containing 0.05% Triton-X-100 (Roth) and a protease inhibitor cocktail (Roche) at 37 °C for 1.5 min to remove the soluble Tubulin. Then, neurons were fixed with cold methanol (−20 °C) for 8 min, permeabilized and blocked with PBSS (PBS containing 0.2% saponin (Serva), and 2.5% BSA, Sigma) for 30 min at RT and incubated with primary antibodies diluted in PBSS for 2 h (see Supplementary Table1). Coverslips were rinsed twice in PBSS (2 min each) and incubated with corresponding secondary antibodies (diluted 1:500) in PBSS for 30 min. Finally,

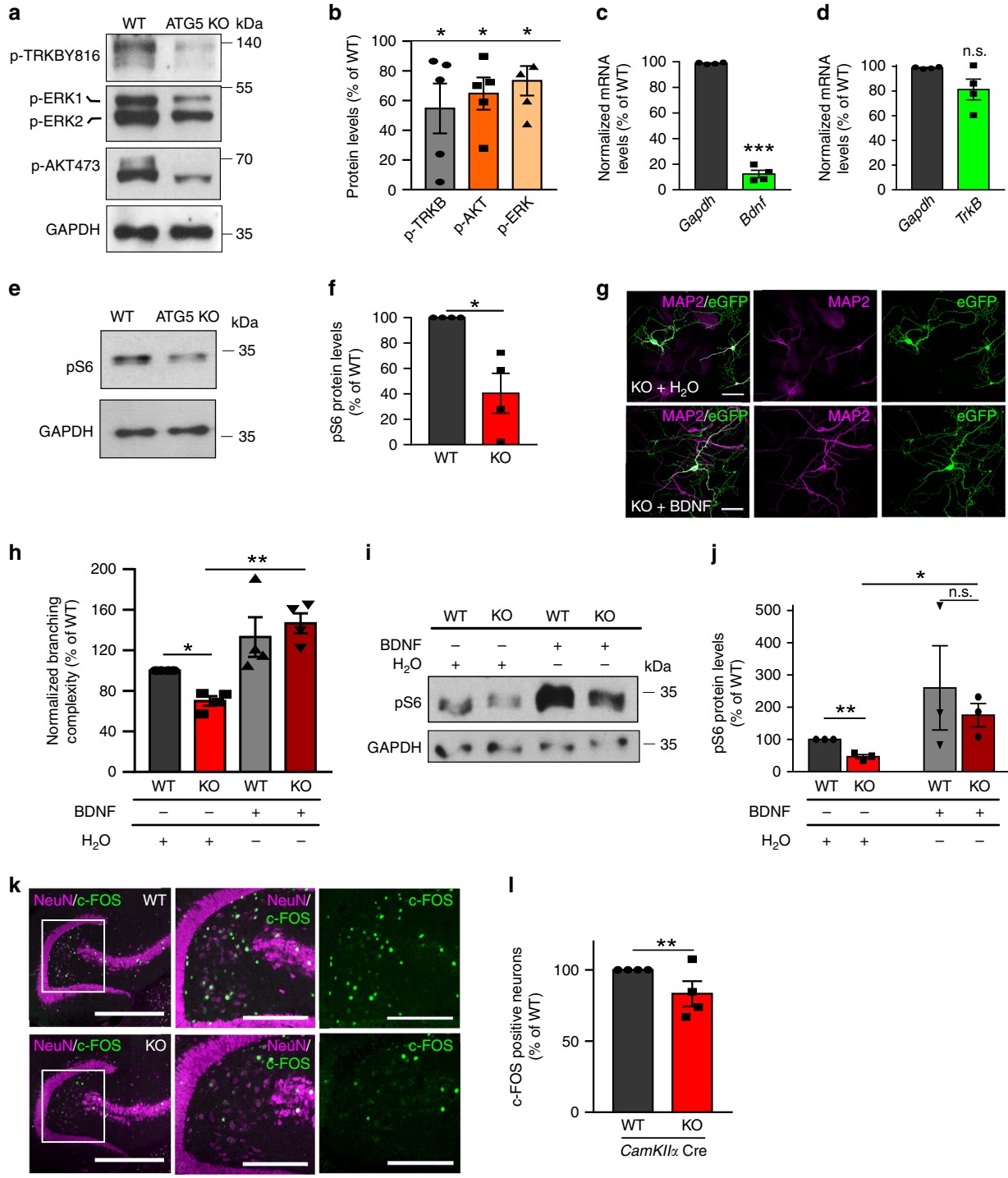

coverslips were washed four times in PBSS and mounted in Immu-Mount (Thermo Scientific).

**STED imaging**. Cultured cortical/hippocampal neurons were fixed under MTs stabilization conditions. In brief, a 3-min extraction at RT in PEM buffer (2% polyethylene glycol ($M_r$ 10,000), 100 mM PIPES-KOH, pH 6.9, 1 mM MgCl$_2$, 1 mM EGTA) containing 1% Triton X-100 (Thermo Scientific) and 2 μM taxol was followed by a quick rinse in PBS before fixation with 3.7% PFA and 0.5% glutaraldehyde in PEM buffer for 20 min. Subsequent MT extraction of neurons was performed by using 0.5% Triton X-100 (Roth), 5% BSA in PBS, followed by antibody incubation without further blocking reagents or detergents. Coverslips were mounted in ProLong Gold (Thermo Fisher Scientific). STED imaging with time-gated detection was performed using a commercial SP8 gSTED ×3 microscope (Leica Microsystems), equipped with a pulsed excitation white light laser

(WLL; B80-ps pulse width, 80-MHz repetition rate; NKT Photonics) and two STED lasers for depletion (continuous wave at 592 nm, pulsed at 775 nm). The pulsed 775-nm STED laser was triggered by the WLL. Within each independent experiment, samples were acquired with equal settings. Alexa 488 and Alexa 532 were excited using a pulsed WLL at 488 and 545 nm, respectively. Depletion occurred at 592 nm. Fluorescence signals were detected sequentially by hybrid detectors at appropriate spectral regions separated from the STED laser by corresponding dichroic filters. Images were acquired with an HC PL APO CS2 ×100/1.40-N.A. oil objective (Leica Microsystems), a scanning format of 1024 × 1024, eight-bit sampling, and 4.5 zoom, yielding a pixel dimension of 25.25 nm and 25.25 nm in the x and y dimensions, respectively.

**Sholl analysis of cultures neurons**. Neurons were transfected with eGFP on DIV 8 and fixed on DIV 16-18 were imaged with Leica SP8 confocal microscope using a

**Fig. 7 Loss of MT dynamics in ATG5 KO neurons impairs neurotrophin signaling and compromises the learning and memory-dependent neuronal activation. a, b** TRKB signaling is significantly decreased in lysates from cultured ATG5 KO neurons compared to the WT set to 100% (KO$^{p\text{-}TRKB}$: 54.68 ± 16.78%, *$p$ = 0.027; KO$^{p\text{-}ERK}$: 73.34 ± 9.94%, *$p$ = 0.028; KO$^{p\text{-}Akt}$: 64.70 ± 10.82%, *$p$ = 0.016). $N$ = 5 independent experiments. **c** *Bdnf* mRNA levels are significantly decreased in cultured ATG5 KO neurons (KO$^{Bdnf}$: 12.35 ± 2.88%, KO$^{Gapdh}$: 98.79 ± 0.36%). ***$p$ < 0.000. $N$ = 4 independent experiments. **d** *TrkB* mRNA levels are non-altered in ATG5 KO neurons (KO$^{TrkB}$: 81.25 ± 8.37%, KO$^{Gapdh}$: 98.79 ± 0.36%). **e, f** Protein levels of phosphorylated S6 Ribosomal Protein (pS6) in lysates from cultured WT and ATG5 KO neurons (KO: 40.39 ± 14.04%). *$p$ = 0.016. $N$ = 4 independent experiments. **g** ATG5 KO neurons expressing eGFP were treated either with BDNF or with $H_2O$ and subsequently immunostained for MAP2. Scale bar, 50 μm. **g, h** Application of BDNF rescues the branching complexity of ATG5 KO neurons compared to $H_2O$-treated KO controls (KO$^{BDNF}$: 146.55 ± 9.88, KO$^{H_2O}$: 69.98 ± 4.72, **$p$ = 0.001). $N$ = 4 independent experiments. See also Supplementary Fig. 7 for detailed Sholl analysis. **i, j** pS6 protein levels are significantly increased in lysates of ATG5 KO neurons treated with BDNF (KO: 175.16 ± 36.20%), comparing to $H_2O$-treated KO controls (KO: 46.61 ± 6.40%). *$p$ = 0.049. $N$ = 3 independent experiments. **k** Representative confocal images of c-FOS labeled nuclei (green) in the dentate gyrus in WT and ATG5 KO mice sacrificed after acquiring the memory on novel object recognition. To reveal the neurons, sections were co-immunostained for NeuN (magenta). Scale bars, 800 μm, 200 μm inserts. **l** The number of c-FOS-labeled nuclei is significantly decreased in the dentate gyrus of mice lacking ATG5 (75.12 ± 8.88%) when compared to the WT. *$p$ = 0.020. $N$ = 4 independent experiments. All graphs show mean ± SEM, statistical analysis was performed by unpaired two-tailed Student's $t$-test in (**f, m, s, t, u, v, l**), paired Student's $t$-test in (**h**) and one-sample Student's $t$-test in (**b, c, d, f, j**). In all graphs the KO was normalized to the WT set to 100%. n.s.-non-significant. Total number of neurons in $N$ experiments is shown in Supplementary Table 3. Source data are provided as a Source Data file.

Plan-Apochromat ×63/1.32 Oil DIC objective, corrected for both chromatic and spherical aberrations and analyzed as previously described[35]. In brief, Sholl analysis of single neurons was performed using the ImageJ Sholl Analysis Macro. The complexity of neuronal branching was calculated by summing the number of intersections within 200 μm from the cell body.

**Live imaging of cultured neurons**. Neurons were imaged on DIV 16-19 (or at DIV 6 in Supplementary Fig. 2x) at 37 °C in imaging buffer (170 mM NaCl, 3.5 mM KCl, 0.4 mM $KH_2PO_4$, 20 mM N-Tris[hydroxyl-methyl]-methyl-2-ami-noethane-sulphonicacid (TES), 5 mM $NaHCO_3$, 5 mM glucose, 1.2 mM $Na_2SO_4$, 1.2 mM $MgCl_2$, 1.3 mM $CaCl_2$, pH 7.4) using Zeiss Axiovert 200 M microscope (Observer. Z1, Zeiss, USA) equipped with 63x/1.40 Oil DIC objective, a pE-4000 LED light source (CoolLED) and a Hammatsu Orca-Flash4.O V2 CMOS digital camera. Time-lapse images of neurons expressing TRKB-mRFP were acquired every second using Micro-Manager software (Micro-Manager1.4, USA) for 30 s. In the case of experiments with EB3-Tdtomato, Mito-Cherry, eGFP-CLASP2, eGFP-LC3B, eGFP-LC3BG120A, eGFP-GABARAPG116A plasmids videos were acquired every second for 60 s. Kymographs were generated using the software Kymo-Maker[67] and analyzed by ImageJ. To analyze the number of EB3 comets, small axonal fragments were selected from individual neurons and the number of comets/μm was quantified manually from the kymographs. To monitor the exo-cytosis of synaptic vesicles via SYB2-pHluorin assay, neurons were subjected to electrical field stimulation at 50 Hz for 4 s using an RC-47FSLP stimulation chamber (Warner Instruments) and imaged as described above (time-lapse images were acquired every second for 2 min). To prevent the activation of postsynaptic receptors, 10 μM CNQX and 50 μM AP-5 were added to the imaging buffer prior to imaging. To determine the specificity of MT based transport of TRKB-mRFP carrier vesicles, DIV15-16 neurons were treated with 20 μm Ciliobrevin D (EMD Millipore Corp, USA) for 3 h prior to the live cell imaging.

Live and dead neurons were analyzed using LIVE/DEAD™ Cell Imaging Kit (Thermo Fisher Scientifics). Following manufacture indications, an equal amount of imaging buffer was mixed with the dye and added to WT and ATG5 KO neurons (DIV 16) for 15 min at RT. Subsequently, neurons were imaged using Zeiss Axiovert 200 M microscope (Observer. Z1, Zeiss, USA) equipped with ×20 objective, a pE-4000 LED light source (CoolLED) and a Hammatsu Orca-Flash4.O V2 CMOS digital camera. The analysis was performed using ImageJ.

**Synthesis of cell-permeable LC3B peptide**. LC3B peptides corresponding to the amino acid sequence 108-125 (GeneID: 67443) were attached N-terminally to a cell-penetrating peptide (CPP) sequence, namely sC18*[68]. For control, LC3B peptide sequence was scrambled and fused to the CPP. Peptides were synthesized on Wang resin (loading was 15 mmol) using an automated solid phase peptide synthesizer (MultiSyntech, Syro I), and following standard Fmoc/tBu-strategy. For each coupling step, 7 equivalents of Fmoc-amino acid and 7 equivalents of activation reagents (ethyl-2-cyano-2-(hydroxyimino)acetate (Oxyma) and N,N'-diiso-propylcarbodiimide (DIC)) were used. N-terminal labeling with 5,6-carboxytetramethylrhodamine (Tamra) was achieved by using 3 equivalents of Tamra and 3 equivalents Oxyma and DIC, respectively. All reactions were per-formed in DMF (dimethylformamide). Following, peptides were cleaved by incu-bating the resin for 3 h with a mixture of concentrated trifluoroacetic acid (TFA)/1,2-ethanedithiol (95/5 (v/v)). Subsequently, peptides were precipitated in ice-cold diethyl ether, washed and lyophilized. All peptides were purified using preparative reversed-phase (RP) HPLC (Hitachi Elite LaChrom (VWR), RP18 column Nucleodur C18ec, 100–5 (Macherey-Nagel)) at 6 mL min$^{-1}$ and 220 nm detection. Acetonitrile/water with 0.1% TFA were used as eluents, changing gradient as needed. Fractions were pooled and lyophilized. Identity was proven by LC-ESI-MS

(RP-HPLC from Agilent coupled to an ESI LTQ XL (Thermo) using a Chromolith Performance RP-18e column, 100 × 4.6 mm from Merck, and using a gradient of 10–60% acetonitrile (0.1% formic acid) in water (0.08% formic acid) over 15 min and a flow rate of 0.6 mL min$^{-1}$. Purities were >95%. LC3B (108–125): (Tamra-GLRKRLRKFRNKFLYMVYASQETFGTAMAV): 3997.2 Da (calc.), 3997.4 (exp.)
Scr: (Tamra-GLRKRLRKFRNKGFESMAAYVMTVYQALFT): 3997.2 Da (calc.), 3997.4 (exp.).

**Analysis of axonal swellings in GFP-LC3B transgenic neurons**. Cortical and hippocampal neurons were isolated from postnatal GFP-LC3B transgenic mice at p1–5 as previously described above. Neurons were treated at DIV15 with 5 μM of LC3B or scrambled peptide for 3 h at 37 °C. Following treatment neurons were washed with HBSS medium and further incubated in the medium for 15 h. Live cell images were acquired using Zeiss Axiovert 200 M microscope with ×10 objective lens. Total number of neurons was quantified from each image and swellings were represented as a percentage of total neurons.

**Electron microscopy**. Neurons cultured on 18 mm Ø coverslips were fixed with pre-warmed fixative solution (2% glutaraldehyde, 2.5% sucrose, 3 mM $CaCl_2$, 100 mM HEPES, pH 7.4) at RT for 30 min, followed by the post-fixation at 4 °C for 30 min. Afterwards, the cells were washed with 0.1 M sodium cacodylate buffer, incubated with 1% $OsO_4$, 1.25% sucrose, 10 mg/ml $K_4[Fe(CN)_6] \cdot 3\ H_2O$ in 0.1 M cacodylate buffer for 1 h on ice and washed three times with 0.1 M cacodylate buffer. Subsequently, the cells were dehydrated at 4 °C using ascending ethanol series (50, 70, 90, 100%, 7 min each), incubated with ascending EPON series (EPON with ethanol (1 + 1) for 1 h; EPON with ethanol (3 + 1) for 2 h; EPON alone ON; 2 × 2 h with fresh EPON at RT) and finally embedded for 48–72 h at 62 °C. Coverslips were removed with liquid nitrogen and heat, consecutively. Ultrathin sections of 70 nm were made using an ultramicrotome (Leica, UC7) and stained with uranyl acetate for 15 min at 37 °C and lead nitrate solution for 4 min. Electron micrographs were taken with a JEM-2100 Plus Electron Microscope (JEOL), Camera OneView 4 K 16 bit (Gatan) and software DigitalMicrograph (Gatan).

**Novel object recognition test**. The novel object recognition (NOR) was per-formed on 12–13-week-old *Atg5*$^{flox/flox}$:*CamKIIα*-Cre WT and KO mice in a white painted square box (50 × 50 × 40 cm), illuminated with >20 lux. The objects used in the test included a blue colored cool pack and a diverse colored cube. The objects differed in height, width, color and shape and no preference for one of the objects was determined in preliminary experiments. Mice were monitored during the session by a digital camera placed above the arena and connected to a video tracking system (EthoVision®XT, Noldus). Animals were handled for 5 min on three successive days prior to the test session. Habituation to the environment was performed 60 min before the test sessions, exposing the animal to the open field for 5 min. Two training session (60 min interval) were performed with two copies of one object, placed in the back left and right corners of the box, followed by the test session 24 h later with one object used in the training sessions and one novel object. The animals were exposed for 5 min to the objects and each mouse was placed individually at the mid-point of the wall opposite to the objects with its nose pointing away from the objects. Mice were sacrificed via transcardial perfusion (see below) 90 min after the test session.

**Immunohistochemical analysis of brain sections**. Mice were sacrificed at 12–15 weeks-old by an overdose of ketamine/xylazine (i.p., 10 μl per 10 g body weight) and transcardially perfused with 50 ml saline solution (0.85% NaCl, 0.025% KCl, 0.02% $NaHCO_3$, pH 6.9, 0.01% heparin, body temperature), followed by 50 ml cold

(7–15 °C) freshly depolymerized 4% (wt/vol) PFA (Merck) in 0.1 M PBS, pH 7.4. Brains were carefully taken out of the skull, postfixed overnight in the same fixative, and placed in a mixture of 20% (vol/vol) glycerol (Roth) and 2% (vol/vol) dimethylsulfoxide (VWR International) in 0.4 M PB for 24 h for cryoprotection Free-floating sections (40 μm) were made using freezing Microtome KS34 (Microm GmbH) and collected in six series in 0.1 M PBS. For immunostaining, corresponding horizontal 40 μm sections from WT and KO littermates were washed three times in PB (3 × 15 min each), followed by permeabilization with PB containing 0.3% Triton X-100 (9 × 20 min each). Sections were preincubated with PB containing 5% (vol/vol) NGS (Thermo Scientific) and 0.3% Triton X-100 (Roth) for 1 h and subsequently incubated with primary antibody at 4 °C for 48 h (see Supplementary Table1). Then, sections were washed 9 times for 20 min each in 0.3% Triton X-100 in 0.125 M PB and incubated with Alexa-conjugated secondary antibodies (1:500) for 2 h using standard techniques. The sections were washed three times for 15 min each in 0.125 M PB. Finally, sections were mounted on gelatin-coated glass slides. Sections were imaged either at a resolution of 1024 × 1024 (or at 2048 × 2048 in Fig. 6d, e), with eight-bit sampling (no z increment was used) with a Leica SP8 confocal microscope ((Leica Microsystems), using a Plan-Apochromat ×63/1.32 oil DIC objective, corrected for both chromatic and spherical aberrations.

**Nissl staining**. Horizontal brain sections were obtained from perfused 13-week-old ATG5 KO and ATG16L1 KO mice and processed for cresyl violet staining. For that, the sections were mounted in 0.2% gelatine solution in 250 mM Tris-HCl and dried overnight at 40 °C on a heating plate. Mounted sections were re-hydrated for 1 min in water and stained for 5–10 min in 0.1% cresyl violet solution. Subsequently, sections were rinsed three times (2 min each) in water and dehydrated using an ascending ethanol series (50, 70, 80, 90%) for 2 min each. After rinsing the sections in 96% ethanol, they were destained with 0.5% acetic acid and washed twice in 100% ethanol (2 min each), incubated with xylene for 2 min and subsequently mounted using Entellan[35].

**Western blot**. Mice were sacrificed at 12–14 week-old by cervical dislocation. Brains were dissected and immediately placed into liquid nitrogen to be stored at −80 °C for further use or directly homogenized with a Wheaton Potter-Elvehjem Tissue Grinder in radioimmunoprecipitation assay (RIPA) buffer, containing 50 mM Tris pH 8.0, 150 mM NaCl, 1.0% IGEPAL CA-630, 0.5% Sodium deoxycholate, 0.1% SDS and protease inhibitor cocktail (Roche), sonicated and incubated for 1 h on ice for the protein extraction. To extract protein from cortical/hippocampal primary cultures, neurons were harvested at DIV16-18 with RIPA buffer containing protease inhibitor and phosphatase inhibitor cocktail (Thermo Scientific), sonicated, and placed for 30 min on ice. The lysates were centrifuged at 17.0×g at 4 °C and supernatants were collected. Protein concentrations were determined by Bradford assay (Sigma). Starvation was induced by incubating the cells at 37 °C/5% CO2 ON for 16 h in osmolarity-adjusted Earle's Balanced Salt Solution (EBSS, Gibco). In order to study the levels of acetylated α-tubulin, neuronal cultures were harvested with a lysis buffer containing 6 M Urea, 50 mM Tris, 150 mM NaCl, 0.1% SDS, 1% Triton, pH 7.4. Depending on the experiment 2–20 μg of total protein were loaded onto gel and separated by SDS-PAGE, and then transferred to a nitrocellulose or PVDF membrane. Membranes were blocked for 1 h at RT in 5% skim milk (or with BSA) in TBS buffer (20 mM Tris pH = 7.6, 150 mM NaCl) containing 0.1% Tween (TBS-T) and incubated with the primary antibodies overnight at 4 °C (see Supplementary Table1), followed by washing three times (10 min each) with TBS-T. Afterwards membranes were incubated with secondary antibodies diluted in 5% skim milk or BSA in TBS-T buffer for 1.5 h at RT. After the incubation, membranes were washed three times as above and subsequently developed using ECL-based autoradiography film system. The analysis was performed using Image J Analyze Gel plugin. For quantification, protein levels were always first normalized to the loading control and then the levels in the KO were normalized to the WT set to 100%.

**Extraction of MT fractions**. To fractionate soluble and polymerized MTs from cultured neurons, first the lysis Buffer A1 (137 mM NaCl, 20 mM Tris-HCl, 1% Triton X-100 and 10% Glycerol) was added to the cells at DIV 16 at 4 °C for 3 min, plates were gently swirled two to three times, and the supernatant was collected as a soluble fraction[69]. Immediately after, polymerized tubulin was extracted using lysis Buffer B (Buffer A + 1% SDS), which was added to the cells for 1 min and the cells were harvested as polymerized fraction. To isolate dynamic MTs, first the soluble fraction was extracted at DIV 16 with the lysis Buffer A2 containing 137 mM NaCl, 20 mM Tris-HCl and 0.1% digitonin (Sigma) for 10 min at RT. Next, to induce the depolymerization of stable MTs, the neurons were incubated with Buffer A2 at 4 °C for 1 h. The supernatant was collected and saved as dynamic MT fraction. Cold-stable MTs were extracted with Buffer B (Buffer A2 + 1% SDS). Collected MT fractions were briefly sonicated and analyzed by Western blotting as described above.

**Co-immunoprecipitation assays**. To immunoprecipitate α-tubulin from cultured WT and ATG5 KO neurons, first soluble MT fraction was extracted as described above (under "Extraction of MT fractions"). Remaining cells were then incubated

with the Co-IP buffer (50 mM Tris-HCl, 1% NP-40, 100 mM NaCl, 2 mM MgCl2, protease inhibitor (Roche), pH 7.4) and the extraction of polymerized tubulin were performed by incubating the cell lysates at 4 °C for 45 min. Following incubation, the lysates were centrifuged at 0.9 × g for 5 min at 4 °C and supernatants were collected and used as polymerized fractions for immunoprecipitation (IP) assay. For the IP Dynabeads Protein G (Invitrogen) were coupled either to α-tubulin antibody or to equivalent amount of non-specific IgG as a control. After the coupling, the unbound antibody was removed by carefully placing the tubes on the magnetic bar. Antibody-coupled Dynabeads were then incubated with soluble and polymerized tubulin fractions. For the IP from the soluble fraction the Dynabeads were incubated overnight at 4 °C to avoid the polymerization of tubulin, while the polymerized fraction was incubated for 1 h at RT. Samples were then washed 3 times using Co-IP buffer. Proteins were eluted in 4xSDS sample buffer and analyzed by Western blotting. For other Co-IP experiments, cultured WT and ATG5 KO neurons (DIV 15-16) or whole mouse brain were homogenized and extracted in Co-IP buffer. Equal amount of cell lysates was added to the Dynabeads Protein G coupled to the target antibody or the non-specific IgG as a control. The samples were further processed as above.

For experiments in HEK293T cells, cells were maintained in DMEM medium (GIBCO), containing 10% FCS, penicillin (255 Units/ml) and streptomycin (255 μg/ml). Twenty-four hours after seeding, the cells were transfected with eGFP or eGFP-LC3B, eGFP-LC3BG120A, eGFP-LC3A, eGFP-LC3A G120A, eGFP-GABARAP or eGFP-GABARAP G116A along with tdTomato-ELKS1. Following 20 h of overexpression cells were harvested and lysed with the RIPA buffer and immunoprecipitation assays were performed by incubating the cell lysates with magnetic GFP-microbeads (Miltenyibiotech, Germany). The samples were analyzed by WB as described above.

**Knockdown experiments in MEF or NSC-34 cells**. Mouse embryonic fibroblasts (MEFs) or mouse motor neuron-like hybrid cell line (NSC-34) were maintained in DMEM medium (GIBCO), containing 10% FCS, penicillin (255 Units/ml) and streptomycin (255 μg/ml) (in case of MEF cells, 1% MEM NEAA (Gibco) were supplemented to the medium). The cells were transfected with 300 pM siRNA using Lipofectamine RNAiMAX Reagent (Thermo Fisher). Gene silencing was achieved by two consecutive rounds of transfections with the interval of 24 h. 48 h after the first transfection cells were harvested and cell lysates were then analyzed by WB.

**Cyclohexamide and MG132 treatment in HEK293T and NSC34 cells**. For the cycloheximide assay, HEK293T cells were transfected with eGFP, eGFP-LC3B or eGFP-LC3BG120A plasmids. Twelve hours post-transfection cells were incubated with 5 μM of cycloheximide. Following 0, 4, 8, and 24 h of incubation cells were harvested in RIPA buffer and samples were analyzed by WB. For the MG132 assay, HEK293T were treated with 5 μM of MG132 (Sigma) or DMSO (dimethyl sulfoxide) as a control. NSC34 cells were treated with 10 μM of MG132 for 24 h.

**Seahorse assay**. The seahorse assay was performed to measure the OCR (oxygen consumption rate) in ATG5 WT/KO neuronal cultures at DIV 13. Neuronal cultures were prepared as described above and plated in a PDL-coated 96-well XF tissue culture plate (Agilent) at a concentration of 20,000 cells/well. Blank vials were distributed equally over the plate, 4 vials were analyzed per genotype. The XFp sensor cartridges were hydrated overnight with XF calibrant at 37 °C. For the measurement/treatment three port A-C were loaded with 10x concentration of mitochondrial inhibitors as stated by the user guide of the Seahorse XFp Cell Mito Stress Test kit (cat-103010-100; Agilent Technologies). Each treatment lasted for 15 min, including 3 measurement time points. The first episode was used to establish the baseline, before sequential injection of oligomycin (1 μM), FCCP (1 μM), rotenone and antimycin A (both 0.5 μM). Data processing was performed using WAVE software version 2.6.0.

**Mass spectrometry analysis of LC3 binding partners**. Brains of 8 weeks old WT mice were homogenized in 2 ml Co-IP buffer (see above). Brain lysates were stored for 45 min on ice before centrifugation at 17,0 × g for 20 min at 4 °C. The supernatants were collected and the protein concentration was assessed by Bradford. The magnetic Dynabeads (Protein G, Invitrogen) coupled to LC3 or IgG antibody as a control were incubated with 8.8 mg of the supernatant overnight on 4 °C on a shaker. Next day, the beads were washed three times with Co-IP buffer before they were transferred into fresh tubes and re-dissolved in 20 μl Co-IP buffer and 4x SDS buffer. After boiling the samples at 95 °C for 5 min the samples were loaded onto SDS-PAGE gels, reduced (DTT) and alkylated (CAA). Digestion was performed using trypsin at 37 °C overnight. Peptides were extracted and purified using Stagetips. Eluted peptides were dried in vacuo, resuspended in 1% formic acid/4% acetonitrile and stored at −20 °C prior MS measurement. All samples were analyzed on a Q-Exactive Plus (Thermo Scientific) mass spectrometer that was coupled to an EASY nLC 1000 UPLC (Thermo Scientific). Peptides were loaded with solvent A (0.1% formic acid in water) onto an in-house packed analytical column (50 cm × 75 μm I.D., filled with 2.7 μm Poroshell EC120 C18, Agilent). Peptides were chromatographically separated at a constant flow rate of 250 nl/min using the following gradient: 5–27% solvent B (0.1% formic acid in 80% acetonitrile) within

40 min, 27–50% solvent B within 8 min, followed by washing and column equilibration. The mass spectrometer was operated in data-dependent acquisition mode. The MS1 survey scan was acquired from 300–1750 $m/z$ at a resolution of 70,000. The top 10 most abundant peptides were isolated within a 1.8 Da window and subjected to HCD fragmentation at a normalized collision energy of 27%. The AGC target was set to 5e5 charges, allowing a maximum injection time of 110 ms. Product ions were detected in the Orbitrap at a resolution of 17,500. Precursors were dynamically excluded for 10 s. All mass spectrometric raw data were processed with Maxquant (version 1.5.3.8) using default parameters. Briefly, MS2 spectra were searched against the Uniprot MOUSE.fasta database, including a list of common contaminants. False discovery rates on protein and PSM level were estimated by the target-decoy approach to 1% (Protein FDR) and 1% (PSM FDR) respectively. The minimal peptide length was set to 7 amino acids and carbamidomethyolation at cysteine residues was considered as a fixed modification. Oxidation (M) and Acetyl (Protein N-term) were included as variable modifications. The match-between runs option was enabled. LFQ quantification was enabled using default settings. The Maxquant output was processed using the Perseus computational platform. Protein groups flagged as „reverse", „potential contaminant" or „only identified by site" were removed from the proteinGroups.txt. Statistical analysis of log2-transformed LFQ values was performed using a one-sided $t$-test. Correction for multiple testing was achieved by a permutation based FDR-approach. The mass spectrometry proteomics data have been deposited to the ProteomeXchange Consortium via the PRIDE (PubMed ID: 26527722) partner repository with the dataset identifier PXD011279 [https://www.ebi.ac.uk/pride/archive?keyword=PXD011279].

**Quantitative RT-PCR**. RNA isolation was performed using Trizol (Thermo Fisher Scientific)[35]. Twenty nanogram of total RNA was used for reverse transcription using the High Capacity cDNA Reverse Transcription Kit (Applied Biosystems) following the manufacturer´s instructions. qPCR was performed with qPCRBIO SyGreen Mix Kit (Applied Biosystems) in a 7500 Fast System. *Gapdh* housekeeping expression was evaluated using two other housekeeping genes, *Rrn18s* and *Hprt1*. qPCR analysis of the mRNA expression of pro-survival and pro-apoptotic genes was performed using a universal PCR master mix and TaqMan probes (Applied Biosystems; Bcl2l1/Bcl-xL: Mm00437783_m1, cflar/cFlipL: Mm01255580_m1, Dtit3/Chop: Mm00492097_m1, Bax: Mm00432051_m1) and the Ct values were normalized to TATA-box-binding protein (*Tbp*) that was used as reference gene. Relative expression of gene transcripts was assessed using the 2-ΔΔCt method[70].

**In-vitro pull-down**. Full-length human LC3B was expressed from pRSF-Duet-1 (G2P) as His$_6$-tagged fusion protein. The expression was performed in *Escherichia coli* BL21 (DE3). The cells were harvested by centrifugation (15 min, 1.5 xg, 4 °C), resuspended in buffer A (100 mM NaCl, 50 mM Tris/HCl pH 7.4, 5 mM MgCl$_2$, 2 mM β-mercaptoethanol, 100 µM Pefabloc) and lysed by sonication. The His$_6$-LC3B containing supernatant after centrifugation (45 min, 33,600 × $g$) was loaded onto an equilibrated Ni-NTA column (Ni Sepharose 6 Fast Flow, GE Healthcare) using buffer B (100 mM NaCl, 50 mM Tris/HCl pH 7.4, 5 mM MgCl$_2$, 2 mM β-mercaptoethanol) plus 20 mM imidazole. The column was washed with 10 column volumes of washing buffer C (300 mM NaCl, 50 mM Tris/HCl pH 7.4, 5 mM MgCl$_2$, 20 mM imidazole, 2 mM β-mercaptoethanol) and the His$_6$-LC3 protein eluted using a 20 mM to 500 mM Imidazole gradient in buffer B. The His$_6$-LC3B containing fractions were pooled and concentrated using a 3 kDa MWCO Amicon ultrafiltration unit for the subsequent size-exclusion chromatography on a HiLoad 26/600 Superdex 75 pg column (GE Healthcare) in buffer D (50 mM HEPES pH 7.9, 100 mM NaCl, 5 mM MgCl$_2$, 2 mM β-mercaptoethanol, Roth). The His$_6$-LC3 containing fractions were pooled, shock-frozen in liquid nitrogen and stored at −80 °C. Purified recombinant MYC/DDK tagged human ERC1 protein was purchased from (OriGene CAT#: TP313864). For the assay 2 µg of His$_6$-LC3B was mixed with 1 µg of MYC/DDK-ERC1 in reaction buffer (50 mM Tris, pH 7.6, 500 mM NaCl, 5 mM MgCl$_2$, 2 mM β-mercaptoethanol, 10% glycerol, 30 mM Imidazole) with 30 µl of Ni-NTA magnetic agarose beads (Qiagen). For the control group only MYC/DDK-ECR1 was added to the Ni-NTA magnetic beads. The reaction mixture was incubated for 1 h at 4 °C with gentle shaking. Following the incubation, the beads were washed three times with the reaction buffer and proteins were eluted from the beads by adding 4× SDS-PAGE sample buffer and the samples were analyzed by immunoblotting using ELKS1 and LC3B antibodies.

**Statistical analysis and reproducibility**. For analysis of experiments, statistically significant estimates were obtained from at least three independent experiments (indicated as N in figure legends), which represent biological replicates. The exact number of neurons used for single independent experiments in indicated in Supplementary Table 3. Graphs and statistical analysis was made with GraphPad Prism (GraphPad Software 8.0.1). Statistical tests used are given in the figure legend of the corresponding experiment. The statistical significance between two groups for all normally distributed raw data except growth factor treatments was evaluated with a two-tailed unpaired student's $t$ test. Effect of BDNF on neuronal complexity were evaluated using paired student's t tests. The statistical significance between more than two groups for all normally distributed raw data was evaluated using either One-Way ANOVA with Dunnett's posthoc test or Two-Way ANOVA repeated

measures with Tukey posthoc test for multiple comparisons. All normalized data were evaluated using one-sample student's $t$ test. Statistical significance on variables obtained from Sholl analysis was calculated using two-way ANOVA with repeated measurements after data normalization using Ln transformation. Significant differences were accepted at $p < 0.05$. All shown representative WB images or micrographs are examples from minimum $N = 2$ independent experiments.

**Supplemental information**. Supplemental Information includes Supplementary Tables 1–3 and 8 supplementary figures and can be found with this article online.

**Reporting summary**. Further information on research design is available in the Nature Research Reporting Summary linked to this article.

## Data availability
All data supporting the findings described in this study are available from the corresponding author upon reasonable request. Source data from all representative blots shown in this study as well as data underlying all quantitative analysis performed in this study are provided with the manuscript as a Source Data file. Uncropped WB membranes for representative WB images are shown in Supplementary Fig. 8. Mass spectrometry data have been deposited in the ProteomeXchange Consortium via the PRIDE (PubMed ID: 26527722) partner repository with the dataset identifier PXD011279.

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

## Acknowledgements

We thank N. Ellrich and M. Schäfer for expert technical assistance. The work of N.L.K. is funded by the Deutsche Forschungsgemeinschaft (DFG, German Research Foundation) under Germany's Excellence Strategy – CECAD (EXC 2030 – 390661388), DFG (KO5091/2-1) and Fritz Thyssen Foundation (Az. 10.18.1.036MN). A.N. is a member of the RTG-NCA, DFG 233886668/GRK1960. The lab of MK is funded from DFG (KY96/1-1). We thank Dr. N. Brose and Dr. H. Kawabe (MPI of Exp. Medicine, Göttingen) for providing the tdTomato-ELKS1 plasmid, Dr. M. Kreutz (LIN, Magdeburg, Germany) for providing the pmCherrry-N1 plasmid, Prof. M. Bergami (CECAD, Cologne, Germany) for providing the Ai9(RCL-tdT) line, Prof. A. Trifunovic (CECAD, Cologne, Germany) for providing the CamKIIα-Cre line and Prof. P. Rosenstiel (University Hospital Kiel, Germany) for sharing the ATG16L1 flox/flox mice. We thank Prof. Noboru Mizushima

(Univ. of Tokyo) for providing the GFP-LC3 transgenic mice. We are indebted to CECAD Proteomics and Imaging Facilities for expert assistance, to Dr. T. Sesia (University Hospital of Cologne) for the help in using the Noldus Software, to Dr. T. MacVicar (MPI for Aging Research) for the help with Seahorse assay, to M. Drexelius for the help in peptide synthesis, to Prof. O. Gruß (University of Bonn) for the help in designing MT fractionation experiments and to Dr. M. Graef (MPI for Ageing Research), Prof. B. Wirth (University Hospital Cologne), and Prof. T. Langer (MPI for Aging Research) for critical comments on the manuscript.

## Author contributions

A.N.-H., M.O., and N.L.K. designed experiments. A.N.-H., M.O., S.B., E.D., K.S., M.K., C.Q., M.L., V.K, I.N., and N.L.K performed experiments. A.N.-H. and N.L.K. wrote the manuscript with the input from other coauthors.

## Competing interests

The authors declare no competing interests.
