## [Peer Review File · Nature Communications]

Reviewers' comments:

Reviewer #1 (Remarks to the Author):

The manuscript by Negrette-Hurtado et al uncovers a potentially interesting property of the autophagic lipidation machinery in regulating the stability of microtubules that seems to be independent of the function of this lipidation machinery in autophagy.

The work is based on the observation that ATG5 and ATG16 knockout excitatory neurons show the expected accumulation of the adaptor p62 (SQSTM1) in the soma but not in the axons, although it is in the axons of these animals that strong swellings and degeneration are seen. Importantly, these phenotypes are not seen in FIP200 KO neurons. Further experiments showed that in the KO neurons several endocytosis-related defects were seen leading to the idea that absence of the lipidation machinery may affect microtubule-dependent trafficking. To provide a molecular explanation for this, the authors show that the LC3 protein interacts with protein ELKS1 and CLASP2, and in this way it affects microtubule stability. Thus, the model presented suggests that in the ATG5 and ATG16 KO neurons the presence of unlipidated LC3 impairs microtubule stability and axonal transport, leading to autophagy-independent pathologies.

Although I was very impressed with the thoroughness of this investigation, I think that the authors may have the wrong suspect (unlipidated LC3) in custody for a few reasons.

1. Cells, including neurons, can tolerate large increases in LC3 without showing neurodegeneration defects. Although the wild type LC3 would flux towards the lipidated form, at any given time there would be plenty of the unlipidated form in these overexpressing cells presumably able to cause damage. Of note, there is a GFP-LC3 mouse with very high expression in neurons. Also, the increase in the levels of LC3 type I that the authors see in their animals is very modest.
2. Binding of LC3 to the microtubule regulators is extremely modest, and it increases almost imperceptibly in the knockouts. If this were the basis of the phenotype, I would expect a much larger increase in the binding of the KOs.
3. Rescue of the phenotypes using the G120A LC3 mutant, and the binding of the various regulators to this mutant must be shown very extensively throughout the paper. This is potentially the strongest result linking the phenotype to unlipidated LC3. Along those lines, the inlipidated GABARAP could be used as a negative control.

An alternative explanation for the data in this manuscript has to do with the fact that in the absence of the lipidation machinery cells will still accumulate all sorts of early membranous autophagosomal structures that may interfere with a variety of cellular processes. Although this is overlooked in some publications, the data are extensive and very strong. See for example these recent papers:

<https://mcb.asm.org/content/34/9/1695/article-info>

<https://www.molbiolcell.org/doi/full/10.1091/mbc.e08-03-0309>

<http://jcs.biologists.org/content/127/18/4089>

Interestingly, these structures do not accumulate in KOs of early autophagy genes such as FIP200 for example because in these cells the pathway is blocked at the earliest stage.

In terms of the neuronal events, and given recent work from Holzbaur and colleagues showing dynamics of autophagosome formation in axons, it is very likely that some of these pre-autophagosomal structures accumulate (and possibly damage) axons. It is then worth investigating whether the presence of these structures may explain the endosomal effects seen, which in turn could relate to the microtubule changes. I very strongly urge the authors to look into this possibility. Markers for these early stalled intermediates include WIPI2, FIP200, ATG13 and ATG9. Antibodies are available for all.

Minor comments

Levels of LC3 type I and type II inconsistent between experiments (Fig 1b vs 2I)

Is total tubulin reduced in the KOs?

Please add loading controls throughout

In the GABARAP vs LC3 binding experiment, input for GABARAP is very low, making interpretation difficult. There is some non-specific stickiness in those binding experiments in general which, coupled to the modest effects overall, makes me sceptical that this is the right interpretation.

N. Ktistakis

Reviewer #2 (Remarks to the Author):

In this manuscript by Negrete-Hurtado et al, the authors report a macroautophagy-independent role of LC3 in the regulation of microtubule (MT) dynamics in neurons via a newly discovered direct interaction with active zone protein ELKS1, which acts via the MT-binding protein CLASP2. This mechanism appears to be pathophysiologically relevant in that failure to properly regulate MT dynamics results in dystrophic axons, a feature that is present in multiple neurodegenerative disorders.

This is an exciting study that reports an unexpected role for LC3 in the control of MT dynamics via a novel protein interactor. The study is well designed, well controlled and includes important mechanistic aspects. There are only a few minor aspects that need to be addressed to make it suitable for publication in Nature Comms, in this referee's view. These are as follows:

- It is unclear how LC3 controls the stability of ELKS1 and CLASP2. Is it by preventing proteasomal degradation or chaperone-mediated autophagy? Is LC3 affecting the half life of those proteins? Figuring it out would add value to the manuscript.

- The abstract emphasizes how important autophagy is in controlling MT dynamics, but the main message is that LC3 acts in an autophagy-independent fashion. The impact of autophagy is therefore

largely indirect. This should be made more clear in the abstract, otherwise it adds confusion and misrepresents the study a bit.

- The figure legends should mention whether sample sizes reflect technical or biological replicates. Currently, it is unclear how many independent experiments were performed, particularly for cell culture experiments.

Reviewer #3 (Remarks to the Author):

In this manuscript, Kononenko and colleagues studied the cytoskeletal changes in the autophagy mutants. Their primary observation was that axonal varicosities form in the several autophagy mutants. They did a good job to show that these defects are not due to degradation defects but instead likely caused by the accumulation of non-lipidated cytosolic LC3. They further showed that in the ATG5 mutants, retrograde trafficking of TrkB vesicles are defective and that MT stability is increased. They provide strong evidence that LC3 associates with ELKS1 and speculated that ELKS1's association with CLASP, an +TIP protein was responsible for the cytoskeleton changes. Overall, this manuscript contains a large number of experiments. The first part of the paper was very good. However, the major mechanistic part that characterizing the MT changes, trafficking defects, and ELKS1 are too preliminary. The mechanistic links are rather weak, which diminishes the quality of this paper. For example, ELKS1 is a presynaptic active zone protein. None of the staining showed an active zone like staining pattern. The link between the over-stable MT and defective retrograde trafficking is not clear at all. No experiments answered the question how CLASP2 causes the trafficking defects. The combination of technical concerns and the weak mechanistic data reduces my enthusiasms towards this paper.

Major comments:

1. Is the change of RAB7 specific to axonal varicosities? How about RAB7 staining in soma and cell body.
2. The conclusion of Fig. 3l and m about dynactin localization is not valid. Dync1 looks cytosolic in both wild type and mutant. Similarly in 3n and O, it is difficult to appreciate that there are enrichment of TrkB in the varicosity. It is likely that the presence of TrkB in the varicosity is due to the large volume and the presence of membrane organelles there as they have shown with the EM.
3. The typical dynein mediated axonal retrograde movement of TrkB vesicles is around 2 $\mu\text{m}/\text{s}$. the speed of control particles reported in Fig. 3 is 0.3 $\mu\text{m}/\text{s}$. This make me worry if they are actually looking at Dynein mediated transport.

4. Fig. 5 g needs to be repeated. The amount of the ELKS protein is very low in the control lane. In addition, the LC3 bands seem to be much more intense compared to the ELKS band. Since this is a binding experiment performed on purified proteins, a commassie blue gel should be shown to provide an estimate of stoichiometry.
5. why is the ELKS1 staining in Fig. 6d not synaptic at all? Does the antibody recognize ELKS1?
6. the hypothesis that increased CLASP2 is responsible for the MT changes needs to be directly demonstrated.

Minor comments:

1. Is the expression level of LC3 and LC3(G120A) similar to each other in Fig. 2q? How is spheroids defined?
2. Line 198 "We found that autophagy-deficient synapses indeed accumulate late endosomal organelles (Fig. 3i) and functional mitochondria .
3. Normal synapses also contain mitochondria. If the authors intend to suggest that the mitochondria is increased in the ATG5 mutants, they should quantify this. How do they know the mitochondria is functional?
4. CLASP2 antibody also needs to be validated for immunostaining experiments.

CECAD Cologne • Joseph-Stelzmann-Str. 26 • 50931 Cologne

To
Nature Communications
The Macmillan Campus, 4 Crinan Street, London N1 9XW, United Kingdom

December 3rd, 2019

CECAD Cluster of Excellence
University of Cologne

CECAD Research Center
Joseph-Stelzmann-Str. 26
50931 Cologne | Germany

Dr. Natalia L. Kononenko

www.cecad.uni-koeln.de

Reply to the reviewers

We would like to thank the reviewers for their time and effort into careful reading and commenting of our work via their comments, suggestions, scrutiny as well as positive feedback on our manuscript. Their constructive suggestions have substantially improved our paper. Attached are our detailed point-by-point responses to reviewers.

All changes in the manuscript have been marked with yellow color.
On the behalf of the authors

Yours sincerely,

Natalia Kononenko

Reviewer #1 (Remarks to the Author):

The manuscript by Negrette-Hurtado et al uncovers a potentially interesting property of the autophagic lipidation machinery in regulating the stability of microtubules that seems to be independent of the function of this lipidation machinery in autophagy.

Response: We thank the Reviewer 1 for this positive comment on our work.

The work is based on the observation that ATG5 and ATG16 knockout excitatory neurons show the expected accumulation of the adaptor p62 (SQSTM1) in the soma but not in the axons, although it is in the axons of these animals that strong swellings and degeneration are seen. Importantly, these phenotypes are not seen in FIP200 KO neurons. Further experiments showed that in the KO neurons several endocytosis-related defects were seen leading to the idea that absence of the lipidation machinery may affect microtubule-dependent trafficking. To provide a molecular explanation for this, the authors show that the LC3 protein interacts with protein ELKS1 and CLASP2, and in this way it affects microtubule stability. Thus, the model presented suggests that in the ATG5 and ATG16 KO neurons the presence of unlipidated LC3 impairs microtubule stability and axonal transport, leading to autophagy-independent pathologies.

Although I was very impressed with the thoroughness of this investigation, I think that the authors may have the wrong suspect (unlipidated LC3) in custody for a few reasons.

1. Cells, including neurons, can tolerate large increases in LC3 without showing neurodegeneration defects. Although the wild type LC3 would flux towards the lipidated form, at any given time there would be plenty of the unlipidated form in these overexpressing cells presumably able to cause damage. Of note, there is a GFP-LC3 mouse with very high expression in neurons.

Response: We understand and appreciate the reviewer's concern regarding the absence of neurodegeneration in a GFP-LC3 mouse. We would like to note that in agreement with reviewer's comment, we do not see axonal pathology in cultured neurons transiently overexpressing GFP-LC3 WT (Fig. 2p,q), which indeed indicates that LC3 mostly fluxes towards the lipidated form and at the healthy condition the ratio between LC3I and LC3II is maintained constant. Our hypothesis is that the phenotype is caused by the misbalance of two isoforms, and that the specific accumulation of cytoplasmic LC3 is required to cause the changes in organelle trafficking. To test this hypothesis, we took advantage of the GFP-LC3 transgenic mouse line, which is available in the lab of our colleague Prof. Pasparakis. First, we analyzed GFP-LC3 levels in the brain versus liver samples. We did not find a particular enrichment of GFP-LC3 in neurons comparing to liver samples (**new Fig. S2k**). Second, to cause the misbalance between two isoforms and to stabilize the cytoplasmic LC3 in GFP-LC3 transgenic neurons we synthesized cell permeable peptides corresponding to the LC3B aa108-125, as well as a scramble version, both tagged with TAMRA. Our hypothesis was to stabilize the cytoplasmic LC3 by inducing a dominant-negative effect via titrating the ATG4 in neurons treated with this peptide (Satoo et al., 2009). We treated GFP-LC3 transgenic neurons with these peptides for 3 hours, followed by washing and post-incubation period in conditioned media for 15hours. We found that the number of cells with axonal swellings was significantly increased in GFP-LC3 neurons treated with LC3B peptide (**new Fig. S2l,m**). Taken together, our data indicate that the balance between cytoplasmic and lipidated isoform might be a critical determinant of axonal homeostasis. These data are now incorporated in the revised version of the manuscript, page 8.

Also, the increase in the levels of LC3 type I that the authors see in their animals is very modest.

Response: To better illustrate the increase in LC3-I levels, we have now added 2 more experiments in Fig. 2m (LC3I levels *in-vitro*). Our analysis indicates that the LC3-I levels are app. 1.9-fold increased in ATG5 KO neurons (**new Fig. 2m**). Furthermore, we have now quantified the levels of LC3I in conditional ATG5 KO brains (**new Fig. S1d**). We observed that LC3I levels are also significantly upregulated *in-vivo* (1.3 fold). We also would like to mention that the LC3I levels in Fig.1 come from conditional KO brains, where specifically excitatory neurons lack ATG5 (only around 30% of all neurons, Wang et al., 2013) and, thus, the increase in LC3-I *in-vivo* is underestimated.

2. Binding of LC3 to the microtubule regulators is extremely modest, and it increases almost imperceptibly in the knockouts. If this were the basis of the phenotype, I would expect a much larger increase in the binding of the KOs.

Response: We have now repeated the co-IP experiments in cultured neurons (**new Fig. 6a**). We observe a noticeable increase of LC3 binding to ELKS1 in KO neurons, which scales according to LC3I accumulation observed under this condition.

3. Rescue of the phenotypes using the G120A LC3 mutant, and the binding of the various regulators to this mutant must be shown very extensively throughout the paper. This is potentially the strongest result linking the phenotype to unlipidated LC3. Along those lines, the inlipidated GABARAP could be used as a negative control.

Response: We thank the reviewer for this constructive suggestion. We have gladly followed his suggestion and performed a series of co-IPs, as well imaging experiments using lipidation-deficient LC3A, LC3B and GABARAP constructs. We found that: i) both unlipidated LC3AG120A and LC3BG120A bind ELKS1. Interestingly, we observed a stronger binding of LC3BG120A to ELKS1, comparing to LC3AG120A mutant (**new Fig. 5e,f**); ii) lipidation-deficient GABARAP does not bind ELKS1 (**new Fig. 5g**); iii) in live-imaging experiments we observed that overexpression of unlipidated LC3BG120A, but not LC3B per se or GABARAPG116A, decreases the MT dynamics, monitored by EB3 comet assay (**new Fig. 4q-t**).

4. An alternative explanation for the data in this manuscript has to do with the fact that in the absence of the lipidation machinery cells will still accumulate all sorts of early membranous autophagosomal structures that may interfere with a variety of cellular processes. Although this is overlooked in some publications, the data are extensive and very strong. See for example these recent papers:

<https://mcb.asm.org/content/34/9/1695/article-info>

<https://www.molbiolcell.org/doi/full/10.1091/mbc.e08-03-0309>

<http://jcs.biologists.org/content/127/18/4089>

Interestingly, these structures do not accumulate in KOs of early autophagy genes such as FIP200 for example because in these cells the pathway is blocked at the earliest stage.

In terms of the neuronal events, and given recent work from Holzbaur and colleagues showing dynamics of autophagosome formation in axons, it is very likely that some of these pre-autophagosomal structures accumulate (and possibly damage) axons. It is then worth investigating whether the presence of these structures may explain the endosomal effects seen, which in turn could relate to the microtubule changes. I very strongly urge the authors to look into this possibility.

Markers for these early stalled intermediates include WIPI2, FIP200, ATG13 and ATG9. Antibodies are available for all.

Response: We thank the reviewer for this constructive suggestion. We have now performed a detailed investigation of pre-autophagosomal structures in KO axons using antibodies against WIPI2, FIP200, ATG9 and ATG13 (data are from N=3 independent experiments). We have not detected any changes in the intensity/amount of FIP200 and ATG13 in KO neurons (**new Fig. S3h-o**).

Interestingly, we even detected a significant decrease for levels of ATG9 and WIPI2 in the mutant condition, likely due to impaired membrane trafficking from the soma. Additionally, we also provide the WB data illustrating that the levels of WIPI2, FIP200, ATG13 remain largely unchanged in lysates in KO neurons (**new Fig. S3q**). These data argue that axonal swellings are not a result of accumulation of early autophagosome intermediates. We have incorporated these data in the revised version of the manuscript, page 9.

Minor comments

Levels of LC3 type I and type II inconsistent between experiments (Fig 1b vs 2l)

Response: The same point as above. Levels of LC3 in Fig.1 come from conditional KO brains, where specifically excitatory neurons lack ATG5 (only about 30% of all cells in blotted lysates), and, thus, the increase in LC3-I in Fig. 1 is underestimated. In contrast, the lysates blotted in Fig. 2l come from the full ATG5 KO.

Is total tubulin reduced in the KOs?

Response: Total tubulin levels are not altered in neither ATG5 nor ATG16L1 KO neurons. We have incorporated now these data in the **new Fig. S4e-h**.

Please add loading controls throughout

Response: We are assuming that the Reviewer is referring to experiments where the levels of post-translationally modified tubulin were blotted against total tubulin. These experiments are performed using MT fractionation protocol, where stable and dynamic MT fractions are separately extracted and analyzed. The only proper loading control is alpha-tubulin. Since, the total levels of alpha tubulin are not altered (Fig. S4e-h), we are convinced that it can be used as a loading control.

In the GABARAP vs LC3 binding experiment, input for GABARAP is very low, making interpretation difficult. There is some non-specific stickiness in those binding experiments in general which, coupled to the modest effects overall, makes me sceptical that this is the right interpretation.

Response: We have now repeated the co-IPs with GABARAP (**new Fig. 5d**). In N=3 experiments we did not observed the binding.

Reviewer #2 (Remarks to the Author):

In this manuscript by Negrete-Hurtado et al, the authors report a macroautophagy-independent role of LC3 in the regulation of microtubule (MT) dynamics in neurons via a newly discovered direct interaction with active zone protein ELKS1, which acts via the MT-binding protein CLASP2. This

mechanism appears to be pathophysiologically relevant in that failure to properly regulate MT dynamics results in dystrophic axons, a feature that is present in multiple neurodegenerative disorders.

This is an exciting study that reports an unexpected role for LC3 in the control of MT dynamics via a novel protein interactor. The study is well designed, well controlled and includes important mechanistic aspects.

There are only a few minor aspects that need to be addressed to make it suitable for publication in Nature Comms, in this referee's view. These are as follows:

- It is unclear how LC3 controls the stability of ELKS1 and CLASP2. Is it by preventing proteasomal degradation or chaperone-mediated autophagy? Is LC3 affecting the half life of those proteins? Figuring it out would add value to the manuscript.

Response: We thank the Reviewer 2 for his/her kind and encouraging comments on our study. We have gladly followed his/her suggestion and have added a set of experiments where we analyzed the stability of ELKS1 and CLASP2 in neurons and their degradation routes. To shed the light on the mechanism by which LC3 controls ELKS1 protein stability, we first speculated that ELKS1, similarly to its *Drosophila* homologue Bruchpilot, is degraded by the proteasome (Zang, Ali et al. 2013). Indeed, we found that ELKS1 protein levels were upregulated in mouse neuronal cells treated for 24h with a specific proteasome inhibitor MG132 (**new Fig. S6f,g**). Furthermore, in cells overexpressing the LC3BG120A mutant and treated with the MG132 the amount of ELKS1 was significantly higher than that in LC3B wildtype -expressing cells (**new Fig. S6h,i**). By examining ELKS1 protein turnover using cycloheximide as a protein synthesis inhibitor, we found that LC3BG120A increases the stability of ELKS1, suggesting that cytoplasmic LC3B regulates ELKS1 protein stability by preventing its proteasomal degradation (**new Fig. S6k,m**). Interestingly, although CLASP2 was not degraded by the proteasome per se, we found that its protein levels were also stabilized by the LC3BG120A in a proteasome-dependent manner (**new Fig. S6j,l**). Taken together, this data indicate that LC3 controls ELKS1/CLASP2 protein complex stability by preventing proteasomal degradation of ELKS1. We added now this information in the revised version of the manuscript on page 14-16.

- The abstract emphasizes how important autophagy is in controlling MT dynamics, but the main message is that LC3 acts in an autophagy-independent fashion. The impact of autophagy is therefore largely indirect. This should be made more clear in the abstract, otherwise it adds confusion and misrepresents the study a bit.

Response: We thank the reviewer for this suggestion. We have now modified the abstract accordingly, which now reads as following: *“Here we report on **a novel regulation of MT dynamics by AuTophaGy(ATG)-related proteins, which previously have been linked to the autophagy pathway.** We find that ATG proteins required for LC3 lipid conjugation are dispensable for survival of excitatory neurons and instead regulate MT stability via controlling the abundance of the MT-binding protein CLASP2. **This function of ATGs is independent of their role in autophagy and requires the active zone protein ELKS1**”*

- The figure legends should mention whether sample sizes reflect technical or biological replicates. Currently, it is unclear how many independent experiments were performed, particularly for cell culture experiments.

Response: We understand and appreciate the reviewer's concern regarding the sample size. The information provided in the figure legends refers to the biological replicates (Ns) and, additionally contains the number of technical replicates (the number of cells/axons). In the previous version of the manuscript under Statistical analysis we indicated that "*For analysis of experiments, statistically significant estimates were obtained from independent experiments (N)*". We apologize however for not making this information clear enough for the reader and state now in the end of each figure legend the following information: "*All data shown represent the mean \pm SEM from N independent experiments, which represent biological replicates*".

Reviewer #3 (Remarks to the Author):

In this manuscript, Kononenko and colleagues studied the cytoskeletal changes in the autophagy mutants. Their primary observation was that axonal varicosities form in the several autophagy mutants. They did a good job to show that these defects are not due to degradation defects but instead likely caused by the accumulation of non-lipidated cytosolic LC3. They further showed that in the ATG5 mutants, retrograde trafficking of TrkB vesicles are defective and that MT stability is increased. They provide strong evidence that LC3 associates with ELKS1 and speculated that ELKS1's association with CLASP, an +TIP protein was responsible for the cytoskeleton changes. Overall, this manuscript contains a large number of experiments. The first part of the paper was very good.

Response: We thank the Reviewer 3 for his/her positive comment on our manuscript.

However, the major mechanistic part that characterizing the MT changes, trafficking defects, and ELKS1 are too preliminary. The mechanistic links are rather weak, which diminishes the quality of this paper. For example, ELKS1 is a presynaptic active zone protein. None of the staining showed an active zone like staining pattern.

Response: We understand and appreciate the reviewer's concern regarding the localization pattern of ELKS1. Indeed, on our hands ELKS1 is strongly enriched at synaptic active zones (for details please see below). We apologize for not making this clear in the previous version of the manuscript. In the revised version of the manuscript, we have now incorporated the data showing synaptic active zone-like pattern for ELKS1 in the new **Fig. S6b and Fig. 6d,e (see also new Fig. 5m)**. Furthermore, we have also analyzed the enrichment of ELKS1 at the active zone of WT and KO mice in the brain *in-vivo* (**new Fig. 6f**). We observed a significant enrichment of ELKS1 at synaptic active zones in the mutant condition.

The link between the over-stable MT and defective retrograde trafficking is not clear at all. No experiments answered the question how CLASP2 causes the trafficking defects.

Response: We understand and appreciate the reviewer's concern. We have now performed new experiments showing that the upregulation of CLASP2 levels in control neurons is enough to decrease the MT dynamics. We observed that CLASP2 upregulation directly decreases MT catastrophe frequency and increases the length of MT growth event (**new Fig. 6r-v**). These data suggest that CLASP2 protects MTs against depolymerization and reduces the number of dynamic

plus ends. Surprisingly, the mean velocity of EB3-tdTomato comets was not significantly decreased in CLASP2 overexpressing axons (**new Fig. 6v**), which indicates that CLASP2 is not regulating the polymerization rate of the remaining plus-ends. Since MT depolymerization induces the recruitment of dynein to kinetochores (Wordeman, Steuer et al. 1991), and several plus-end tracking proteins, required for retrograde trafficking localize to the ends of tyrosinated microtubules (dynamic), but not to the ends of detyrosinated (stable) MTs (Peris, They et al. 2006), our data suggest that increased CLASP2 levels might block the retrograde cargo loading to the MTs and increase cargo stalling at axonal swellings. This part of the text is now included in the manuscript on the page 16.

Major comments:

1. Is the change of RAB7 specific to axonal varicosities? How about RAB7 staining in soma and cell body.

Response: We have now quantified the levels of RAB7 in the cell body (soma) and found no difference between WT and KO condition. We provide this quantification in **new Fig. S3t**.

2. The conclusion of Fig. 3l and m about dynactin localization is not valid. Dync1 looks cytosolic in both wild type and mutant. Similarly in 3n and O, it is difficult to appreciate that there are enrichment of TrkB in the varicosity. It is likely that the presence of TrkB in the varicosity is due to the large volume and the presence of membrane organelles there as they have shown with the EM.

Response: We agree with the reviewer and thank him/her for this comment. We have now modified the main text and state on page 10 that *“Our data revealed that while in control condition DYNC1 showed a homogeneous cytosolic appearance along the axons, ATG5 KO axons revealed large 5-10µm spheroid-like accumulations of DYNC1”*. Furthermore, to better illustrate this point, we have now included a different snapshot image for WT and KO axons for **Fig. 3l**. We also would like to note that all quantifications of protein levels in the axons are normalized to the area, to account for volume difference between WT and KO.

We also fully agree on the fact that the presence of TrkB in the varicosity might be due to the large volume and the presence of membrane organelles shown in the EM. Our hypothesis is that axonal swellings in the KO contain trapped cargo of MT-based transport. To support this point, we have now provided a Pearson’s correlation coefficient in **Fig. S3u**. We also modified now the text on the page 10, which now says: *“Furthermore, in agreement with the presence of membrane organelles at the KO synapse detected by the EM, we found that KO swellings revealed an increased colocalization of DYNC1 with activated tropomyosin-related kinase receptor B (TRKB) receptors (Fig. 3n,o, Fig. S3u), a known cargo of dynein motors in axons (Kononenko, Claßen et al. 2017)”*.

3. The typical dynein mediated axonal retrograde movement of TrkB vesicles is around 2 µm/s. the speed of control particles reported in Fig. 3 is 0.3µm/s. This make me worry if they are actually looking at Dynein mediated transport.

Response: We thank the reviewer for this constructive comment. Indeed, the yeast cytoplasmic dynein has been described to show the movement with an average run length of ~2 µm (Reck-Peterson et al., 2006). Interestingly, the velocity of porcine brain dynein was reported to be around 800 nm/sec (Toba et al., 2006). Studies with purified mammalian dynein indicate that dynein is a fast motor, with velocities from 0.5 to 1 µm/s (Maday et al., 2014). There are several studies reporting on different kinetics of TrkB vesicles in neurons. For instance, Frederic Saudou and Thomas Willnow

labs reported that the TrkB vesicles in primary striatal and hippocampal neurons move with the speed around 0.6- 0.8 μ m/sec (Liot et al., 2013, Rohe et al., 2013), while the lab of Kimberley McAllister describes the velocity of ca. 0.34 μ m/sec in cortical neurons (Gomes et al., 2006) and the lab of Erica Holzbaaur describes the transport of TrkB with kinetics of 0.6 μ m/sec in DRG neurons (Klinman & Holzbaaur, 2015). Interestingly, neurons imaged around 8-12 DIV (days in vitro) or earlier are reported to exhibit higher TrkB velocities. Thus, we speculated that the velocity of TrkB carriers might be regulated developmentally. To test this hypothesis, we have now performed the analysis of TrkB kinetics in neurons at DIV6 and plotted the results as a histogram together with velocity obtained at DIV16 (the age of neurons used in the current study). This data presented in **new Fig. S3z** reveal that the kinetics of TrkB carriers is faster in DIV6, indicating that the speed of these vesicles might be developmentally regulated.

To demonstrate that TrkB is transported via Dynein motor in the axons, we treated the neurons with ciliobrevin D (20 μ M, Ayloo et al., 2017), which is a cell-permeable benzoyl dihydroquinazolinone derivative that acts as a reversible and specific blocker of AAA+ ATPase motor cytoplasmic dynein (Firestone et al., 2012). We found that ciliobrevin D significantly decreased the percentage of moving TrkB vesicles in the axon (**from 55% to 14%**). We also observed that the remaining motile vesicles exhibited significantly decreased retrograde velocity, suggesting that TrkB trafficking is mediated predominantly by dynein motor in DIV16 cortical axons. We have incorporated these data in **new Fig. S3x,y**.

4. Fig. 5 g needs to be repeated. The amount of the ELKS protein is very low in the control lane. In addition, the LC3 bands seem to be much more intense compared to the ELKS band. Since this is a binding experiment performed on purified proteins, a coomassie blue gel should be shown to provide an estimate of stoichiometry.

Response: We agree with the reviewer that this experiment is not suitable to determine the interaction stoichiometry. However, as judged by the outcome of the pulldown experiment the interaction is most likely of medium affinity (maybe in the micromolar range), with rather high dissociation rate. Therefore, using this pull-down assay and immunoblotting or coomassie blue staining are both not suitable to determine the stoichiometry of the interaction, as the affinity is too low, most likely with a rather high dissociation rate and complex dissociation occurs during washing steps and during the course of the experiment. This would result in an inaccurate value for the stoichiometry.

We were able to load 125 ng of LC3 and 250 ng of ELKS1 protein of the input onto the SDS-PAGE gel for immunoblotting. However, as this is not a stoichiometric interaction as for almost all medium to low affinity pulldowns and immunoprecipitations in literature, we lose some ELKS1 protein during the course of the pulldown experiment due to washing steps. Assuming from the immunoblotting using the ELKS1-antibody, we have an app. >10-fold reduction of ELKS1 in the elution, suggesting the amount of <12.5 ng. This would be beyond the sensitivity of coomassie blue staining. Furthermore, considering the huge molecular weight difference of ELKS1 (app. 130 kDa) and LC3B (app. 15 kDa) a direct comparison via coomassie blue staining would be a difficult task. Our aim in this pulldown experiment was rather to show that ELKS1 and LC3 directly interact. This is from our point of view clearly demonstrated by this experiment. We have also repeated the experiment and included the new example in the **Fig. 5g**. As visible from the input lane, a similar amount of ELKS1 was used for the pull down and LC3 protein is only present in the positive control with ELKS1. The fact that in the eluate after pulldown there is more LC3 present compared to ELKS1

is due to the pulldown performed via Ni-NTA-magnetic beads. Using this affinity enrichment, we enrich for His₆-LC3 explaining the intense band in the eluate.

As mentioned above, in this experiment it is not expectable to observe a 1:1 stoichiometry even if there is a true 1:1 stoichiometry for the interaction. A detection of pulldowns and immunoprecipitations by Western-blotting is a well-accepted readout. A stoichiometry could only be detected using pulldown experiments if the interaction has a very tight affinity (nanomolar-picomolar range) but also a lower affinity can of course have a true 1:1 stoichiometry, i.e. one molecule of protein A binds to one molecule B. Due to a lower affinity/high off-rate not all binding sites are occupied and ELKS1 is removed by washing steps from the equilibrium. As all biochemical reactions, the LC3-ELKS1 interaction is an equilibrium reaction and assuming a medium affinity with a high off-rate does not allow to determine the stoichiometry neither by using immunoblotting nor coomassie blue staining. However, as stated above the aim of these experiments was to show an interaction between ELKS1 and LC3B, which from our point of view is clearly demonstrated by our data.

5. why is the ELKS1 staining in Fig. 6d not synaptic at all? Does the antibody recognize ELKS1?

Response: We understand and appreciate the reviewer's concern. We have now performed series of experiments to prove that the ELKS1 antibody recognize ELKS1 and to illustrate its localization in cultured neurons and brain sections. In **new Fig. S6a** we show that ELKS1 antibody recognizes the overexpressed ELKS1 tagged with tdTomato. Furthermore, we show that ELKS1 localizes to the active zones *in-vitro* (**new Fig. S6b**) and *in-vivo* (**new Fig. 6d-e**), visualized with Bassoon antibody. Furthermore, we now show that ELKS1 localization to the active zones is significantly increased in KO brains (**new Fig. 6f**).

6. the hypothesis that increased CLASP2 is responsible for the MT changes needs to be directly demonstrated.

Response: We understand and appreciate the reviewer's concern. We have now performed new experiments showing that the upregulation of CLASP2 levels in control neurons is enough to decrease the MT dynamics. We observed that CLASP2 upregulation directly decreases MT catastrophe frequency and increases the length of MT growth event (**new Fig. 6r-v**). These data suggest that CLASP2 protects MTs against depolymerization and reduces the number of dynamic plus ends. Surprisingly, the mean velocity of EB3-tdTomato comets was not significantly decreased in CLASP2 overexpressing axons (**Fig. 6v**), which indicates that CLASP2 is not regulating the polymerization rate of the remaining plus-ends. Since MT depolymerization induces the recruitment of dynein to kinetochores (Wordeman, Steuer et al. 1991), and several plus-end tracking proteins, required for retrograde trafficking localize to the ends of tyrosinated microtubules (dynamic), but not to the ends of detyrosinated (stable) MTs (Peris, They et al. 2006), our data suggest that increased CLASP2 levels might block the retrograde cargo loading to the MTs and increase cargo stalling at axonal swellings. This part of the text is now included in the manuscript on the page 16.

Minor comments:

1. Is the expression level of LC3 and LC3(G120A) similar to each other in Fig. 2q? How is spheroids defined?

Response: We have quantified the levels of LC3 and LC3G120A and provide these data in **Fig. S2j**.

We observed that the LC3G120A is expressing slightly less comparing the LC3 wildtype plasmid. For the definition of spheroid, we relied on our data provided in the original Fig. S2e, where the size of axonal swelling with the diameter of 2µm and larger is significantly increased in the KO. Thus, axonal varicosity with a diameter larger than 2µm was defined as an axonal swelling. We apologize for not providing this information in the original manuscript and incorporate this fact in the M&M section on page 24.

2. Line 198 "We found that autophagy-deficient synapses indeed accumulate late endosomal organelles (Fig. 3i) and functional mitochondria. Normal synapses also contain mitochondria. If the authors intend to suggest that the mitochondria is increased in the ATG5 mutants, they should quantify this. How do they know the mitochondria is functional?"

Response: We would like to mention that analysis of mitochondria number has been already shown in old Fig. S3h (now Fig. **S3q,r**). In addition, we have now performed the analysis of mitochondria function in WT and KO neurons using Seahorse assay (measures mitochondrial oxygen consumption rate). We did not detect any difference in mitochondria functionality between WT and KO condition. These data are provided in the **new Fig. S3s**.

3. CLASP2 antibody also needs to be validated for immunostaining experiments.

Response: We understand and appreciate the reviewer's concern. We have now performed series of experiments to validate the CLASP2 antibody. First, we show that CLASP2 antibody recognizes the overexpressed CLASP2 tagged with GFP using immunocytochemistry as a readout (**new Fig. S6t**). Second, we performed siRNA mediated KD of CLASP2 using siRNA smart pools and found that the CLASP2 antibody is detecting the decrease (ca 60%) in CLASP2 protein levels in immunostaining experiments (**new Fig. S6u,v**).

Kononenko, N. L., G. A. Claßen, M. Kuijpers, D. Puchkov, T. Maritzen, A. Tempes, A. R. Malik, A. Skalecka, S. Bera, J. Jaworski and V. Haucke (2017). "Retrograde transport of TrkB-containing autophagosomes via the adaptor AP-2 mediates neuronal complexity and prevents neurodegeneration." *Nature Communications* **8**(14819): 1-16.

Peris, L., M. Thery, J. Fauré, Y. Saoudi, L. Lafanechère, J. K. Chilton, P. Gordon-Weeks, N. Galjart, M. Bornens, L. Wordeman, J. Wehland, A. Andrieux and D. Job (2006). "Tubulin tyrosination is a major factor affecting the recruitment of CAP-Gly proteins at microtubule plus ends." *The Journal of Cell Biology* **174**(6): 839.

Wordeman, L., E. R. Steuer, M. P. Sheetz and T. Mitchison (1991). "Chemical subdomains within the kinetochore domain of isolated CHO mitotic chromosomes." *The Journal of Cell Biology* **114**(2): 285.

Zang, S., Y. O. Ali, K. Ruan and R. G. Zhai (2013). "Nicotinamide mononucleotide adenyltransferase maintains active zone structure by stabilizing Bruchpilot." *EMBO reports* **14**(1): 87-94.

REVIEWERS' COMMENTS:

Reviewer #1 (Remarks to the Author):

The revised version has addressed my original concerns in a satisfying way.

Reviewer #2 (Remarks to the Author):

The authors have addressed all my concerns.

Reviewer #3 (Remarks to the Author):

The authors have performed numerous new experiments to address each of the questions from all three reviewers. The new findings support their model. I have no more concerns.